



# What is the cause(s) of positive ozone trends in three megacity clusters in eastern China during 2015–2020?

Tingting Hu[1], Yu Lin[1], Run Liu[1,2], Yuepeng Xu[1], Boguang Wang[1,2], Yuanhang Zhang[3], Shaw Chen Liu[1,2]

[1]Institute for Environmental and Climate Research, Jinan University, Guangzhou, 511443, China

[2]Guangdong-Hongkong-Macau Joint Laboratory of Collaborative Innovation for Environmental Quality, Guangzhou, 511443, China

[3]State Key Joint Laboratory of Environmental Simulation and Pollution Control, College of Environmental Sciences and Engineering, Peking University, Beijing, 100871, China

*Correspondence to*: Run Liu (liurun@jnu.edu.cn), Shaw Chen Liu (shawliu@jnu.edu.cn)

**Abstract.** Due to a robust emission control policy, significant reductions in major air pollutants, such as $PM_{2.5}$, $SO_2$, $NO_2$, and CO, were observed in China between 2015 to 2020. On the other hand, during the same period, there was a notable increase in ozone ($O_3$) concentrations, making it a prominent air pollutant in eastern China. The annual mean concentration of maximum daily 8-hour average (MDA8) $O_3$ exhibited alarming linear trends of 2.4, 1.1, and 2.0 ppb $yr^{-1}$ in three megacity clusters:

Beijing-Tianjin-Hebei (BTH), Yangtze River Delta (YRD) and Pearl River Delta (PRD). Additionally, there was a significant three-fold increase in the number of $O_3$-exceeding days, defined as MDA8 $O_3$ >75 ppb during the same period. Our analysis indicated that the upward trends in the annual mean concentration of MDA8 were primarily driven by the rise in consecutive $O_3$-exceeding days. Furthermore, from 2015 to 2017, there was a widespread expansion of high $O_3$ concentrations from urban centers to surrounding rural regions, resulting in a more uniform spatial distribution of $O_3$ after 2017. Lastly, we discovered a

close association between $O_3$ episodes featuring four or more consecutive $O_3$-exceeding days and the position and strength of the West Pacific subtropical high (WPSH). The WPSH contributed to meteorological conditions characterized by clear skies, subsiding air motion, high vertical stability in the lower troposphere, increased solar radiation, and positive temperature anomaly at the surface. These favorable meteorological conditions greatly facilitated the formation of $O_3$. Thus, we propose that the worsening $O_3$ trends observed in BTH, YRD and PRD from 2015 to 2020 can be attributed to enhanced photochemical

$O_3$ production resulting from an increased occurrence of meteorological conditions with high solar radiation and positive temperature anomalies under the influence of WPSH and tropical cyclones.

## 1 Introduction

Ozone ($O_3$) is an important greenhouse gas, which can also have adverse effects on human health, vegetation, and materials (Bell et al., 2006; Cohen et al., 2017; Kalabokas et al., 2020; Nuvolone et al., 2018). Surface $O_3$ is a secondary pollutant

produced by photochemical reactions involving $O_3$ precursors such as volatile organic compounds (VOCs), carbon monoxide (CO) and nitrogen oxides (NOx) (Ma et al., 2012; Monks et al., 2015; Wang et al., 2017). Compared to $O_3$ precursors,



meteorological conditions are also crucial factors driving the $O_3$ formation. Solar radiation, temperature, relative humidity, wind speed, and cloud cover have been found to be closely related to $O_3$ formation (Dong et al., 2020; Han et al., 2020; Yin et al., 2019). In addition, large-scale circulations, such as the East Asian monsoon, West Pacific subtropical high (WPSH) and tropical cyclones (TCs) can influence $O_3$ concentration as well (Lu et al., 2019; Rowlinso et al., 2019; Yang et al., 2014; Zhao and Wang, 2017).

The concentrations of air pollutants $SO_2$, NOx, CO, $PM_{10}$ and $PM_{2.5}$ in China have been significantly reduced since 2013 (Li M. et al., 2021; Li et al., 2022; Zhai et al., 2019), thanks to the implementation of "Air Pollution Prevention and Control Action Plan". However, the $O_3$ concentration has dramatically increased and emerged as a major air pollutant in eastern China (Bian et al., 2019; Fu et al., 2019; Wang et al., 2020; Zheng et al., 2018). $O_3$ concentrations are particularly high in the three megacity clusters in eastern China, namely Beijing-Tianjin-Hebei (BTH), Yangtze River Delta (YRD) and Pearl River Delta (PRD) (Gao et al., 2020; Guo et al., 2019; Li K. et al., 2021; Liu et al., 2018; Yang et al., 2019).

Annual mean concentrations of maximum daily 8-hour average (MDA8) $O_3$ in the three megacity clusters are shown in Fig. 1. The linear increasing trends of MDA8 $O_3$ for BTH, YRD and PRD are 2.4, 1.1 and 2.0 ppb $yr^{-1}$, respectively during the period 2015–2020. These trends are unusually large compared to the trends in other parts of China as well as the trends worldwide (Chen et al., 2020; Lu et al., 2018; Professional Committee of Ozone Pollution Control of Chinese Society for Environmental Sciences, 2022; Zhang et al., 2020). Thus, a crucial scientific question is: What is the cause(s) of these large positive trends in $O_3$ concentration? Some recent studies suggested that changing photochemical processes induced by anthropogenic emissions are responsible for these trends (Li et al., 2019; Li et al., 2022; Shao et al., 2021; Wang et al., 2020). However, in our analysis of the $O_3$ trends at individual stations in eastern China during the period 2015–2020, we noticed that the interannual variations of $O_3$ concentration were strongly affected by the position and intensity of WPSH and the presence of TCs in the western Pacific and South China Sea, consistent with the results of a number of recent studies (Chang et al., 2019; Mao et al., 2020; Ouyang et al., 2022; Zhao and Wang, 2017).These results suggest that transport/meteorological parameters associated with WPSH and TCs may play an important role in the large trends of MDA8 $O_3$.

The significant impact of WPSH on weather patterns and $O_3$ concentrations over East China is widely recognized (Bachmann, 2015; Chang et al., 2019; Yin et al., 2019; Zhao and Wang, 2017). It is well established that the WPSH plays a critical role in controlling weather conditions, which in turn affects $O_3$ concentrations. For example, the WPSH is known to contribute to the formation of the East Asian monsoon and influence precipitation patterns in the YRD. Furthermore, it also influences air temperature and precipitation across North and South China (Zhang, 2001; Zhao and Wang, 2017). These changes in meteorological conditions have a profound impact on the photochemical formation, dispersion, and accumulation of $O_3$.

Previous studies have indicated that in the outer regions of TCs, PRD experiences specific atmospheric conditions, e.g., high pressure, low humidity, and intense solar radiation. These conditions often result in consecutive days with elevated levels of $O_3$, as observed in various case studies (Ouyang et al., 2022; Wei et al., 2016). Furthermore, statistical investigations have established several noteworthy connections between TCs and $O_3$ concentrations in the PRD area. For example, the meteorological conditions associated with the TC periphery frequently contributed to the formation of elevated surface $O_3$





levels and aerosols (Deng et al., 2019). In addition, TCs in the East China Sea had a higher likelihood of causing increased $O_3$ concentrations in the PRD region (Zhao et al., 2022). Lastly, TCs in the vicinity of Taiwan, China have the greatest influence on air quality in Hong Kong when compared to TCs in other areas, which is primarily because these TCs facilitate the transportation of air pollutants from the PRD region (Lam et al., 2018).

In this study, we focus on exploring possible contributions to the large positive $O_3$ trends in the three megacity clusters in eastern China by changes in meteorological parameters associated with WPSH and TCs during the period 2015–2020. This paper is organized as follows. In Section 2, the data and methodology used in this study are described. Major characteristics of the $O_3$ interannual variability and trends in the three megacity clusters are discussed in Section 3.1. In Section 3.2, we examine the spatial expansion and quasi-saturation of high $O_3$. The annual change of $O_3$-exceeding days with different durations are also examined. A hypothesis of the cause of $O_3$ trends in three megacity clusters in eastern China during 2015–2020 is presented in Section 3.3. Section 4 presents a summary and conclusions.

## 2 Data and methodology

### 2.1 Pollutant Data

In this study, the observed hourly concentrations of air pollutants, including $O_3$, $NO_2$, CO, $PM_{2.5}$, and $SO_2$ from 2015 to 2020 are obtained from the Chinese National Environmental Ministry of Environmental Protection (http://www.cnemc.cn/en/). Gridded MDA8 $O_3$ data from Tracking Air Pollution in China dataset (http://tapdata.org.cn) with a resolution of 10 km are also used (Xue et al., 2020).

### 2.2 Meteorological Data

The European Centre for Medium-Range Weather Forecasts (ECMWF) Reanalysis v5 (ERA5) dataset (available at https://cds.climate.copernicus.eu/), with a horizontal resolution of $0.25° × 0.25°$ and a time interval of 1 h, was used to analyze the influence of meteorological parameters on $O_3$ pollution. The variables used in this study include 2 m temperature (T2m), surface net solar radiation (SSR). In addition, daily mean relative humidity, geopotential height, zonal and meridional wind at 500 hPa from the National Center for Environmental Prediction (NCEP) and National Center for Atmospheric Research (NCAR) reanalysis (https://psl.noaa.gov/data/gridded/data.ncep.reanalysis.html) at a resolution of $2.5°×2.5°$ are used.

### 2.3 Methods

The Chinese National Ambient Air Quality Standard for MDA8 $O_3$ is 160 μg m$^{-3}$, which corresponds to 75 ppb at 273.15 K and 1 atm. It follows that the $O_3$-exceeding days are defined as MDA8 $O_3$ concentration >75 ppb, while non-$O_3$-exceeding days are defined as MDA8 $O_3$ concentration <75 ppb. According to the duration of $O_3$ pollution, it can be divided into consecutive $O_3$-exceeding days with four or more days ($O_3$ days≥4) and consecutive $O_3$-exceeding days less than four days





(O$_3$ days<4). In addition, some common statistical methods are used in this study, including linear fitting, meteorological synthesis method, and two-tailed Student's t test.

The normalized annual mean O$_3$ concentration of the O$_3$-exceeding days is calculated by adding the O$_3$ concentration of the O$_3$-exceeding day each year and dividing it by the total number of days in the year. The normalized annual mean O$_3$ of the non-O$_3$-exceeding days is calculated by the same method except for the non-O$_3$-exceeding days.

Table 1 lists the criteria and corresponding numbers of low O$_3$ and high O$_3$ stations in the three megacity clusters. Low O$_3$ and high O$_3$ stations are defined basing on the number of O$_3$ exceeding days in 2015. Stations with the number of O$_3$-exceeding days fewer than or equal to the low O$_3$ criterion (second column) are considered as low O$_3$ stations. When more than or equal to the high O$_3$ criterion (4$^{th}$ column), they are considered as high O$_3$ stations. We have tested a few reasonably different criteria and found only some insignificant differences in the results. i.e., the results associated with low O$_3$ and high O$_3$ stations are

robust against reasonable changes in their selection criteria. For example, the relatively large criterion (37 days) of low O$_3$ stations in YRD is intended to include a large enough number of stations (about one third of the total of 152 stations) to be fully representative of low O$_3$ and moderate O$_3$ stations. These results have been compared to those of a more stringent criterion of nineteen days and found no notable change in the major characteristics (Figs. S1 and S2).

### 3 Results and discussion

**3.1 Major characteristics of O$_3$ trends**

Major characteristics of the large positive trends in the annual mean O$_3$ concentration are shown in Figs. 2a, 2b and 2c for BTH, YRD and PRD, respectively, in which the normalized annual mean concentrations of MDA8 O$_3$ in the three megacity clusters are compared to contributions from two groups: The O$_3$-exceeding days and non-O$_3$-exceeding days. The increase in O$_3$-exceeding days is the primary contributor to the substantial increase in the annual mean O$_3$ in all three megacity clusters

from 2015 to 2020. The contribution of O$_3$-exceeding days is affected mostly by the changing number of exceeding days (more than 80%), and secondly but nevertheless significantly by their changes in concentrations (less than 20%) (Tables 2–4). e.g., in BTH the exceeding days were 31, 43, 62, 74, 96 and 78 days in the individual years of 2015–2020, respectively, while their concentrations of those years were 66.42, 64.13, 69.44, 68.21, 70.19 and 69.69, respectively (Table 2 second column). Contributions from non-O$_3$-exceeding days are insignificant (p > 0.1), except that in BTH (Fig. 2a) which shows a significant

declining contribution (p = 0.02) due to the reduced number of non-O$_3$-exceeding days. Therefore, the following discussions on the O$_3$ trends will be focused on the O$_3$-exceeding days.

Annual numbers of single and consecutive O$_3$-exceeding days are shown in Figs. 3a, 3b and 3c for BTH, YRD and PRD, respectively. A drastic two to three-fold increase in the annual numbers of consecutive O$_3$-exceeding days can be seen in all three regions. In contrast, the numbers of single O$_3$-exceeding days show only a slight increase in PRD. These drastic increases

in the annual numbers of consecutive O$_3$-exceeding days are clearly the primary contributors to the trends in O$_3$ shown in Figs.





2a, 2b and 2c. This brings up several key scientific questions: What is the cause(s) of the drastic increases in the numbers of consecutive $O_3$-exceeding days? Is it due to changing $O_3$ photochemical processes or changing meteorological parameters?

## 3.2 Spatial expansion and quasi-saturation of high $O_3$

Another important changing characteristics of $O_3$ concentrations is illustrated in Fig. 4a, which depicts the annual mean
concentrations of MDA8 $O_3$ in BTH during $O_3$-exceeding days for all 78 stations (black line), 14 stations in the highest category of $O_3$ concentration (average 103 ppb) observed in 2015 (red line, denoted high $O_3$ stations hereafter, Table 1) and 13 stations in the lowest category of $O_3$ (average 57 ppb) observed in 2015 (green line, denoted low $O_3$ stations hereafter, Table 1). It is remarkable that $O_3$ concentrations at the low $O_3$ stations caught up within 12 ppb with other stations in merely two years (an increase of about 30 ppb from 2015 to 2017), and actually equaled the average of other stations in 2019. Meanwhile, the high
$O_3$ stations experienced a slight decrease in $O_3$ concentration, albeit not statistically significant. This phenomenon suggests strongly that the annual mean concentrations of MDA8 $O_3$ in BTH experienced a fast (within two years) and widespread spatial expansion of high $O_3$ from urban centers to surrounding regions where $O_3$ concentrations were low in 2015. Temporally most of the expansion was accomplished during 2015–2017. This phenomenon of a fast and widespread expansion of high $O_3$ concentrations from urban centers to surrounding regions were also observed at a slightly less degree in YRD (Fig. 4b) and
PRD (Fig. 4c).

It is worth noting that $O_3$ concentrations at the high $O_3$ stations of approximate 100 ppb in 2015 remained nearly constant or slightly declined throughout the entire period of 2015–2020, while the low $O_3$ stations with $O_3$ concentrations less than about 75 ppb in 2015 in all three megacity clusters experienced significant enhancements in $O_3$ concentration (>5 ppb yr$^{-1}$) during 2015–2017 (Figs. 4a, 4b and 4c). This near-constant $O_3$ phenomenon suggests a quasi-saturation effect of $O_3$ formation when
the annual mean concentration of MDA8 $O_3$ reached approximately 100 ppb.

The expansion and quasi-saturation of $O_3$ raises some interesting scientific questions: What is the cause(s) of the expansion and quasi-saturation? Why did the expansion and quasi-saturation happen mostly during 2015–2017? Did it have anything to do with the increase of consecutive $O_3$-exceeding days as suggested in Fig. 3? These questions are addressed in the following section by examining in detail the spatial expansion of high $O_3$ from urban centers to surrounding regions in BTH, YRD and
PRD during 2015–2017.

The spatial expansion of high $O_3$ from urban centers to surrounding regions in BTH and YRD during 2015–2017 can be clearly visualized in Figs. 5 and 6, respectively. Figures 5a, 5b and 5c show the spatial distribution of daily mean concentrations of MDA8 $O_3$ for $O_3$-exceeding days in BTH in 2015, 2017 and their difference (2017 minus 2015), respectively. Comparing Fig. 5a to 5b, one can see that the area inside the 80-ppb contour (75 ppb is the $O_3$ exceeding standard) expanded by about a factor
of five from 2015 to 2017. The daily average concentration of MDA8 $O_3$ within the BTH box increased from 66.42 ppb in 2015 (31 days, Fig. 5a) to 69.44 in 2017 (62 days, Fig. 5b), which was a difference of 3.02 ppb or a merely 4.5% increase between the two years (Fig. 5c). When accounted for the number of $O_3$-exceeding days, the ratio of MDA8 $O_3$ in all $O_3$-exceeding days between 2017 and 2015 became 2.09. The calculation formula is:



$$(69.44 \times 62)/(66.42 \times 31) = 2.09$$

This implied that the increase in $O_3$ in BTH between 2015 and 2017 shown in Fig. 2a (red line) was almost entirely (95.5%) due to the increase in the number of $O_3$-exceeding days. These results together with those shown in Fig. 3a suggest that the increase in $O_3$ in BTH between 2015 and 2017 was driven primarily by the increase of consecutive $O_3$-exceeding days. Spatially Fig. 5c shows the expansion is mostly to the south and southwest outside of BTH, with YRD getting a lion's share of $O_3$ enhancements. Within the BTH box, the nearly constant concentrations of $O_3$ inside Beijing City (40°N, 116.5°E) coupled

with the southwestward expansion of high $O_3$ in 2017 (Fig. 5c) suggested that there was a quasi-saturation of $O_3$ inside Beijing City, and an expansion of weather systems conducive to $O_3$ formation from Beijing toward the southwest of the BTH box during 2017 (Figs. 5b and 5c). This change of weather conditions also caused significant increases in $O_3$ in YRD and even in southern China as far as the western PRD (Fig. 5c). Nevertheless, the $O_3$ concentrations in YRD and PRD stayed below 70 ppb during the $O_3$-exceeding days of BTH in both 2015 (Fig. 5a) and 2017 (Fig. 5b). In other words, the $O_3$-exceeding days of

YRD and PRD are mostly decoupled (i.e., not occurring at the same time) from those of BTH. A logical explanation for this phenomenon is that the atmospheric conditions conducive to high $O_3$ formation in BTH do not overlap significantly with those conditions of YRD and PRD.

Figs. 6a, 6b and 6c are the same as Figs. 5a, 5b and 5c, respectively, except for YRD. Similar to BTH, one can clearly see the expansion of high $O_3$ from the vicinity of Shanghai City (31°N, 121.3°E) in the northwestern direction reaching as far as the

central BTH box during the period 2015–2017 (Figs. 6b and 6c). Comparing Fig. 6a to 6b, one can see that the area inside the 70-ppb contour expanded from Shanghai and vicinity northwestward by more than a factor of five from 2015 to 2017. This expansion was in different direction from the southwestern expansion occurred in BTH (Fig. 5c). Since it is highly unlikely that any change in emissions could result in these different expansions in YRD and BTH, the logical explanation of the expansion in YRD would be that the weather system conducive to $O_3$ formation moved from the vicinity of Shanghai in 2015

(Fig. 6a) northwestward toward western BTH in 2017 (Figs. 6b and 6c). We note, however, this movement of the weather system does not necessarily mean the direct transport of high $O_3$ or its precursors from the vicinity of Shanghai to central BTH. In fact, the presence of separate rather than contiguous red patches of high $O_3$ (>70 ppb) in southern BTH and northern YRD in Fig. 6b is a clear indication that the high $O_3$ are primarily controlled by local photochemical production from local $O_3$ precursors under the expanded conducive weather conditions, rather than the direct upwind-downwind transport of high $O_3$ or

its precursors. The daily average MDA8 $O_3$ in the YRD box increased from 53.79 ppb in 2015 (31 days, Fig. 6a) to 64.35 in 2017 (40 days, Fig. 6b), which was a difference of 10.56 ppb or a 20% increase between the two years (Fig. 6c). When accounted for the number of $O_3$-exceeding days, the ratio of MDA8 $O_3$ in all $O_3$-exceeding days between 2017 and 2015 became:

$$(64.35 \times 40)/(53.79 \times 31) = 1.54 \ (+54\%)$$

This implied that the increase in $O_3$ in YRD between 2015 and 2017 shown in Fig. 2b (red line) was due to both the increases in $O_3$ concentrations (+20%) and the number of $O_3$-exceeding days (+34%).



Figs. 7a, 7b and 7c are the same as Figs. 5a, 5b and 5c, respectively, except they are for PRD. Unlike BTH and YRD, there was only a slight expansion of high $O_3$ within the PRD box toward the southwest in 2017 compared to 2015 (Fig. 7c). Nevertheless, outside the PRD box there was an extensive expansion of high $O_3$ in eastern China, substantially greater than

the expansion within the PRD box (Fig. 7c). The daily average concentration of MDA8 $O_3$ within the PRD box increased from 61.16 ppb in 2015 (14 days, Fig. 7a) to 65.18 in 2017 (36 days, Fig. 7b), which was a difference of 4.02 ppb or a merely 6.6% increase between the two years (Fig. 7c). After accounting for the number of $O_3$-exceeding days, the ratio of MDA8 $O_3$ in all $O_3$-exceeding days between 2017 and 2015 became 2.74. The calculation formula is:

$$(65.18 \times 36)/(61.16 \times 14) = 2.74$$

This implied that the increase in $O_3$ in PRD between 2015 and 2017 shown in Fig. 2c (red line) was almost entirely (93.4%) due to the increase in the number of $O_3$-exceeding days.

Figures 7a and 7b reconfirm that $O_3$-exceeding days in PRD were mostly decoupled from those in BTH (Figs. 5a and 5b) and YRD (Figs. 6a and 6b), as their spatial distributions were characterized by highly distinctive regional features in both 2015 and 2017. These differences suggest that the $O_3$-exceeding days mostly occur in different days in the three individual regions.

On the other hand, a comparison of Figs. 7c, 6c and 5c reveals a striking common feature of high values in southwestern BTH and northwestern YRD, and low values in eastern parts of all three BTH, YRD and PRD boxes. These common features suggest that the difference between 2015 and 2017 in all three individual regions is likely caused by a common mechanism/process that changed from 2015 to 2017. Moreover, as suggested in Fig. 3, this common mechanism/process must be closely related to higher number of consecutive $O_3$-exceeding days in 2017 over those of 2015.

More evidence against the emissions of air pollutants as a major cause of the expansion and quasi-saturation can be seen in Fig. 8, in which the annual mean concentrations of MDA8 $O_3$ during $O_3$-exceeding days are compared to those of Ox ($O_3+NO_2$) as well as other air pollutants in BTH in 2015–2020. The nearly 30 ppb increases in $O_3$ (Fig. 8a) at low $O_3$ stations from 2015 to 2017 occurred also in Ox (Fig. 8b), suggesting that titration by NO or emission of NO was not the cause of the increases in $O_3$ in 2015–2017, even though the titration effect may well be the cause of the smaller overall trend in $O_3$ during much longer

period 2006–2019 as suggested convincingly by Li et al. (2022). In addition, $PM_{2.5}$ concentrations at high $O_3$ stations in Fig. 8c decreased significantly more than those at low $O_3$ stations from 2015 to 2017, yet the low $O_3$ stations experienced a near 30 ppb increase in $O_3$, while $O_3$ remained essentially constant at the high $O_3$ stations, suggesting that the proposed removal of $HO_2$ radicals by $PM_{2.5}$ (Li K. et al., 2021; Shao et al., 2021) was also not a likely cause of the increases in $O_3$ in 2015–2017. In this context, it should be pointed out that we do not doubt the validity of $HO_2$ removal by $PM_{2.5}$, but its effect was obviously too small to impact on the $O_3$ trend in 2015–2017. Finally, neither CO nor $NO_2$ showed any notable change at low $O_3$ stations

between 2015 and 2017, implying negligible change in $O_3$ precursors, NOx and VOCs, as their emission rates tended to be proportional to those of $NO_2$ and CO, respectively. This again supported the notion that changes in the emissions of $O_3$ precursors were unlikely to be the driving cause of the increases in $O_3$ at low $O_3$ stations from 2015 to 2017.

Comparison of Figure 8a to Figure 1 reveals an interesting point: While the yearly average MDA8 $O_3$ concentrations at all

stations in BTH (green line in Figure 1) shows a significant positive $O_3$ trend of 2.38 ppb yr$^{-1}$ with p=0.01, the black line in





Fig. 8a (MDA8 $O_3$ of all stations during $O_3$ exceeding days) shows an insignificant increasing trend of 1.22 ppb $yr^{-1}$ with p=0.2. This is because the values in Fig. 8 are those of $O_3$ exceeding days, of which $O_3$ concentrations at high $O_3$ stations (red line in Fig. 8a) have a small decreasing trend (albeit insignificant) due to the quasi-saturation effect discussed above. This decreasing trend is the main contributor to the high p value of 0.2 of the black line in Fig. 8a (all stations).

**3.3 Cause(s) of the expansion and quasi-saturation**

Major findings of subsections 3.1 and 3.2 can be summarized as follows: (1) Trends in $O_3$ observed in the three megacity clusters in eastern China during 2015–2020 (Fig. 1) were mainly caused by the large trends of approximately two to three-fold increase in the number of consecutive $O_3$-exceeding days (Figs. 2 and 3). (2) A fast and widespread expansion of high $O_3$ from urban centers to surrounding regions was observed in the three megacity clusters during 2015–2019 (Fig. 4); and the majority

of the expansions were accomplished during the 2015–2017 period (green lines in Fig. 4). (3) The expansions of high $O_3$ in the three megacity clusters were accompanied by a quasi-saturation effect that $O_3$ concentrations at the high $O_3$ stations (high $O_3$ in 2015) of approximate 100 ppb in 2015 remained nearly constant throughout the entire period of 2015–2020, while the low $O_3$ stations (low $O_3$ in 2015) with $O_3$ of about 75 ppb in all three megacity clusters in 2015 experienced significant enhancements in $O_3$ (>5 ppb $yr^{-1}$) during 2015–2017 (Figs. 4a, 4b and 4c). And (4) There is independent evidence, including

spatial distribution of the expansion (Figs. 5 and 6) and inter-annual variations in $O_3$, $O_x$, $NO_2$, CO and $PM_{2.5}$ (Fig. 8), suggesting that transport/meteorology rather than emissions of $O_3$ precursors would more likely be the major cause of the expansion and quasi-saturation. In the following, we explore the evidence in support of changing meteorological parameters as a cause of $O_3$ trends in 2015–2020.

**3.3.1 Changes in meteorological parameters as a cause of $O_3$ trends in 2015–2020**

While a specific process/mechanism has yet to be found as the primary contributor to the trends in $O_3$ observed in the three megacity clusters, the findings summarized above suggest that an examination into transport/meteorological processes involved in $O_3$ episodes with consecutive $O_3$-exceeding days could provide useful information on the identity of the primary contributor. Using BTH as an example, we address this issue in the following by dividing $O_3$ episodes of a given year into two groups: the first group has four or more consecutive $O_3$-exceeding days (labeled $O_3$ days≥4), the second group has less than

four consecutive $O_3$-exceeding days (labeled $O_3$ days<4). Figure 9a shows the mean daily $O_3$ concentrations of the first group in 2015 (mean concentration of 71.14 ppb inside the BTH box, 7 days), Figure 9b shows the mean daily $O_3$ concentrations of the second group (65.04 ppb, 24 days), and Fig. 9c is the difference between the two groups (6.10 ppb, Table 2). Figures 9d–9f are the same as Figs. 9a–9c, respectively, except for 2017. The first group in 2017 had 28 days and mean $O_3$ of 74.43 ppb inside the BTH box, while the second group had 34 days and 65.32 ppb (Table 2). One of the most remarkable differences

between 2017 and 2015 in Figs. 9a–9f was the large number of days with four or more consecutive $O_3$-exceeding days (first group) in 2017 (28 days, Fig. 9d) over that of 2015 (7 days, Fig. 9a), which alone contributed to about 62% of the difference in $O_3$ between 2017 and 2015 as shown in Fig. 2a (red line). Approximately 30% was contributed by the 10 days' difference



(2017 vs. 2015) in the number of days with less than four consecutive $O_3$-exceeding days (second group). The contribution by the higher average concentration of MDA8 $O_3$ of the first group in 2017 is only about 8% (Table 2). These values of

contributions reconfirm what is shown in Fig. 3a, i.e., the greater frequency of episodes with four or more consecutive $O_3$-exceeding days contributes the majority (62%) to the higher $O_3$ in BTH in 2017 vs. 2015, the greater intensity/concentration of $O_3$ during the episodes contributes only about 8%, consistent with the expansion and quasi-saturation effect discussed earlier. The phenomenon of frequency over intensity is even more pronounced when the data of 2015 (4th row and 4th column in Table 2) are compared to those of 2019 (8th row and 4th column in Table 2), in which the higher frequency of the first group of 2019

contributes as much as 83% to the higher $O_3$ in BTH in 2019 vs. 2015.

The phenomena illustrated in Figs. 9a–9f also exist in YRD and PRD as well as in most other years. Figures equivalent to Figs. 9a–9c for all years in the three city clusters (except PRD during 2015–2016, in which no episode with four or more consecutive $O_3$-exceeding days occurred) are provided in the Supplementary Material (Figs. S3–S5). Essential information derived from those figures is summarized in Tables 2–4. The 4th column of Table 2 shows that the number of days with four or more

consecutive $O_3$-exceeding days in BTH increased consistently from 7 days in 2015 to 66 days 2019 and dropped back to 38 days in 2020; this pattern of changes matched very well with those in Fig. 2a (red line). The same can be said for YRD (Table 3) and PRD (Table 4), except there are some minor contributions from the third column in Tables 3 and 4, i.e., days with less than four consecutive $O_3$-exceeding days. Another remarkable point is that the difference between (>=4days) and (<4days) (5th column) in Tables 2–4 is slightly positive (mostly by a few percent) for all three city clusters in all years, which again

implies expansion and quasi-saturation of high $O_3$ in episodes with four or more consecutive $O_3$-exceeding days. In summary, Tables 2–4 show quantitatively that the temporal and spatial changes in $O_3$ concentrations in three megacity clusters of eastern China during 2015–2020 can be mostly attributed to the changes in the number of days with four or more consecutive $O_3$-exceeding days. It follows then that the critical question of our quest for the cause(s) of the remarkable large upward linear trend in $O_3$ of the three megacity clusters becomes: what process/mechanism is conducive to the formation of $O_3$ episodes with

four or more consecutive $O_3$-exceeding days?

In Figs. 10a and 10b the values of SSR and T2m of the episodes with four or more consecutive $O_3$-exceeding days are compared to those of $O_3$ episodes with less than four consecutive $O_3$-exceeding days, and to those of clean days (non-$O_3$-exceeding days). As expected, the $O_3$ episodes with four or more consecutive $O_3$-exceeding days consistently have the highest values of SSR and T2m, while the clean days have the lowest values. This is the case in nearly all years studied as shown in the Supplementary

Material (Fig. S6) and is also generally true in YRD and PRD (Figs. S7 and S8). Coupling the higher values of SSR and T2m in the $O_3$ episodes with four or more consecutive $O_3$-exceeding days depicted in Fig. 10 and greater number of days in the $O_3$ episodes with four or more consecutive $O_3$-exceeding days shown in Fig. 3, we therefore propose a hypothesis as follows: the cause of worsening $O_3$ trends in BTH, YRD and PRD from 2015 to 2020 could be attributed to enhanced photochemical $O_3$ production due to the increased occurrence of meteorological conditions of high solar radiation and positive temperature

anomaly at the surface.



Quantitatively the coupling of Fig. 10 with Fig. 3 can be performed by multiplying the difference between the red (four or more consecutive $O_3$-exceeding days) and green (clean days) values of SSR/T2m in Fig. 10 with the frequency of occurrence (in percentage of total days) of $O_3$ episodes with four or more consecutive $O_3$-exceeding days from Fig. 3. The results are compared to the yearly total $O_3$-exceeding days in Fig. 11. Correlation between the yearly $O_3$-exceeding days and weighed

SSR is very good with R values 0.88 or greater in all three regions, lending strong support for our hypothesis. Correlation between the yearly $O_3$-exceeding days and weighed T2m is high correlated in BTH but not correlated in YRD and PRD, which probably suggests that T2m is not as strongly coupled to $O_3$ formation as SSR. Inclusion of $O_3$ episodes with less than four consecutive $O_3$-exceeding days in Fig. 11 did not change the correction coefficients significantly, supporting the robustness of results shown in Fig.11.

The presence of TCs in the northwest Pacific, specific positions and strengths of WPSH in different regions, and mid-high latitude wave activities can contributed to the increased frequency of meteorological conditions characterized by high solar radiation and positive temperature anomalies at the surface (Hu W. et al., 2023; Mao et al., 2020; Ouyang et al., 2022). The contributions of TCs and WPSH are discussed in the following two sections.

### 3.3.2 Contribution of tropical cyclones in PRD

In collaboration with this study, Hu W. et al. (2023) conducted a statistical analysis to assess the processes that contribute to high $O_3$ formation in PRD when TCs occur in the northwest Pacific. They investigated the impact of the distance between TCs in the northwest Pacific and PRD on the $O_3$ concentration in the PRD from 2006 to 2020. Their findings revealed that the most common process leading to consecutive $O_3$-exceeding days, which are responsible for elevated $O_3$ concentrations, was associated with downdraft and stable atmospheric conditions. These conditions were most frequently observed when TCs were

located in the mid-distance category (700 km–4000 km) from PRD. Interestingly, TCs in this mid-distance category, occurring 49% of the time during the warm season (June–November), accounted for a substantial 80.7% of consecutive $O_3$-exceeding days, which was four times higher than the combined consecutive $O_3$-exceeding days from the other three categories. As shown in Fig. 3 and Section 3.1 that the interannual variations and trends in the annual mean $O_3$ concentrations in PRD between 2015 and 2020 were primarily influenced by the occurrence of consecutive $O_3$-exceeding days. Consequently, it can be inferred that

TCs in the mid-distance category played a critical role in shaping the interannual variations and trends of $O_3$ in PRD during the same time period. Quantitatively, Hu W. et al. (2023) found that the downdrafts associated with mid-distance category TCs accounted for 35 consecutive $O_3$-exceeding days in 2019, compared to only 10–25 days in other years. Interestingly, the large number of consecutive $O_3$-exceeding days in 2019 was primarily attributed to the increased occurrence of downdrafts and stable atmospheric conditions brought about by mid-distance category TCs, rather than the number of TCs, which only showed

a slight increase compared to other years. These results imply that the heightened frequency of downdrafts and stable atmospheric conditions associated with mid-distance category TCs could be the main driver behind the elevated number of consecutive $O_3$-exceeding days in 2019, contributing to approximately 80% of the overall trend in $O_3$ concentrations between 2015 and 2020 (Fig. 3c and black line, Fig.4c). In summary of Section 3.3.1, the mid-distance category TCs could contribute





to about 80% of the overall trend of $O_3$ in PRD in 2015–2020. Less contributions of TCs to the $O_3$ trends in YRD and BTH

are expected because of stronger influence of prevailing westerlies at higher latitudes. Nevertheless, similar evaluations for these regions would be highly valuable.

### 3.3.3 Contribution of WPSH

Mao et al. (2020) made a comprehensive study of an 11-day $O_3$ episode in BTH in 2017 and found it was dominated by the presence of the WPSH and mid-high latitude wave activities. Depending on the position and intensity, WPSH is well known

to be a crucial factor affecting $O_3$ concentrations in various parts of eastern China (Chang et al., 2019; Yin et al., 2019; Zhao and Wang, 2017). During this 11-day $O_3$ episode, the ridge line of WPSH maintained at approximately 22°N from June 24 to June 29, which in combination with mid-high latitude wave activities induced meteorological conditions highly conducive to the $O_3$ production in BTH and northern YRD (Mao et al., 2020).

Following the analysis of Mao et al. (2020), the impact of WPSH on $O_3$ in BTH in April–September has been analyzed in Fig.

12 which depicts the composite 500 hPa geopotential height contours, humidity and winds in BTH in April–September for $O_3$-exceeding days in 2015 (a), clean days in 2015 (b), $O_3$-exceeding days in 2017 (c), clean days in 2017 (d), $O_3$-exceeding days in 2019 (e) and clean days in 2019 (f). The three years 2015, 2017 and 2019 are chosen because their differences in $O_3$ contribute predominately to the overall $O_3$ trends (Figs. 1–2). The importance of WPSH is clearly visible in all Figs. 12a–12f when the 5880 and 5900 gpm isolines (green lines) of $O_3$-exceeding days are compared to those of clean days. In all three

years, the WPSH of the former ($O_3$-exceeding days) were significantly stronger than the latter (clean days) as evident by the strong anticyclonic winds and/or the larger areas inside the 5880 gpm isolines. Even in the case of 2017 when the area inside 5880 gpm isolines of the former looked to be similar to that of the latter, the appearance of 5900 line in the former indicated a stronger WPSH. The strong anticyclonic winds in the former (Figs. 12a, 12c and 12e) force moist air of South China Sea northward into southern China and contributed to extensive clouds and precipitation and thus low $O_3$ formation over southern

China and southern YRD. This difference in the $O_3$ formation between BTH and southern China provides a good explanation to why the $O_3$-exceeding days mostly occur in different time periods in the three megacity clusters as discussed in Section 3.2. Furthermore, over East China Sea the prevailing westerlies were forced northward, slowed down and lead to meteorological conditions in BTH and northern YRD characterized by cloudless sky, sinking motion and high vertical stability in the lower troposphere, as well as high SSR and positive T2m anomaly at the surface. These meteorological conditions were highly

conducive to the formation and accumulation of $O_3$. In contrast, the weaker WPSH of the latter allowed relatively strong westerlies to prevail over BTH during clean days in the three years, which tended to disperse $O_3$ (Figs. 12b, 12d and 12f). e.g., the average wind speed over BTH was about 10 m s$^{-1}$ in Fig. 12b, while only about 5 m s$^{-1}$ in Fig. 12a. Quantitatively Fig. 12c had 31 more $O_3$-exceeding days (93 ppb) than Fig. 12a, the 31 days came at the expense of clean days (52 ppb) (Figs. 12b and 12d). The contribution of these 31 days to the difference in MDA8 $O_3$ between 2017 and 2015 (6.5 ppb, Fig. 1) can be

calculated as follows:

$$((93\times62)+(56\times121))/(62+121)–((86\times31)+(52\times152))/(31+152) = 10.8 \text{ ppb}$$





This difference of 10.8 ppb in MDA8 $O_3$ between 2017 and 2015 was for the period of April to September. It should be divided by 2 and became 5.4 ppb for the yearly difference in MDA8 $O_3$ between 2017 and 2015. This value of 5.4 ppb accounted for 83% of the observed difference in MDA8 $O_3$ between 2017 and 2015 (6.5 ppb, Fig. 1). Similar statement can be made for the

difference in MDA8 $O_3$ between 2019 and 2017 (Figs. 12e and 12c, Fig. 1).

We have made the same analysis for other years as well as for YRD and PRD. The results are mostly similar, and thus presented in the Supplementary material (Figs. S9, S10 and S11). Figs.13a–13d for PRD in 2017 and 2019 are shown because there were interesting anticyclonic circulations over PRD during $O_3$-exceeding days in both years (Figs. 13a and 13c). The 2017 anticyclone was a direct product of the WPSH as it resided within the western tip of the 5880 gpm isoline. The 2019 anticyclone

was also likely associated with the WPSH as the center of anticyclone resided just beneath the 5860 gpm isoline to the west of PRD. The anticyclonic circulations were accompanied by stable downdrafts, low winds, and cloudless sky conditions (short arrows and blue shades in Figs. 13a and 13c), which were highly conducive to the $O_3$ formation. Cloudless sky conditions also occurred in YRD and BTH in Figs. 13a and 13c, but the high wind speed prevented the accumulation of $O_3$. This difference in $O_3$ accumulation between PRD and other two regions provide another good explanation to why the $O_3$-exceeding days mostly

occur in different days in the three megacity clusters as discussed in Section 3.2. Quantitatively Fig. 13c had 27 more $O_3$-exceeding days (90 ppb) than Fig. 13a, the 27 days came at the expense of clean days (39 ppb) (Figs. 13b and 13d). The contribution of these 27 days to the difference in MDA8 $O_3$ in PRD between 2019 and 2017 (6.0 ppb, Fig. 1) can be calculated as follows:

$$((90×62)+(46×182))/(62+182)–((85×35)+(39×209))/(35+209)=11.6 \text{ ppb}$$

This difference of 11.6 ppb in MDA8 $O_3$ between 2019 and 2017 was for the period of April to November. It should be divided by 365/244 and became 7.75 ppb for the yearly difference in MDA8 $O_3$ between 2019 and 2017. This value of 7.75 ppb was 1.75 ppb more than the observed difference in MDA8 $O_3$ between 2019 and 2017 (6.3 ppb, Fig. 1), suggesting a reduction of about 3.5 ppb in MDA8 $O_3$ in the cold months of January–March and December between 2017 and 2019, which was approximately confirmed by the observed reduction of 3.88 ppb.

The presence of anticyclonic circulations over PRD is in good agreement with the results of Ouyang et al. (2022) and Hu W. et al. (2023). The latter authors suggested that the anticyclonic circulations over PRD were primarily caused by TCs in northwestern Pacific. Nevertheless, it is widely acknowledged that the tracks of TCs in the northwestern Pacific are influenced, at least to some extent, by WPSH (Sun et al., 2015; Wang et al., 2017), making it difficult to separate the roles played by the TCs on the anticyclonic circulations and $O_3$ formation from those of WPSH. Clearly, further investigations is needed to fully

understand the complex relationship among WPSH, TCs and $O_3$. Based on these results, we hypothesize that the increased frequency of these meteorological conditions enabled by the changing intensity and position of WPSH could contribute as a major cause of the positive $O_3$ trends in the three megacity clusters in eastern China during 2015–2020.

### 3.4 Uncertainty and cautionary statements

It is worth noting that the analyses conducted in Sections 3.1–3.3 have predominantly relied on correlation or regression

analysis techniques, which do not imply a cause-and-effect relationship. To establish a cause-and-effect link between the



proposed changes in meteorological parameters and $O_3$ trends, it is necessary to employ a mechanistic model that is based on the proposed causes and can accurately reproduce the observed $O_3$ trend. Until such model reproduction is achieved, all correlation or regression findings should be considered as a potential maximum cause-and-effect relationship (Wu et al., 2022). However, current mechanistic models suffer from significant uncertainties, making it difficult to credibly simulate critical

atmospheric processes that regulate $O_3$ formation. These processes include atmospheric transport parameterizations, the sources and sinks of OH, $HO_2$ and $RO_2$ radicals, and the photochemistry of VOCs and OVOCs.

## 4 Summary and Conclusions

Thanks to a strong emission control policy, major air pollutants in China, including $PM_{2.5}$, $SO_2$, $NO_2$ and CO had shown remarkable reductions during 2015–2020. However, $O_3$ concentration had increased significantly and emerged as a major air

pollutant in eastern China during the same time period. The annual mean concentration of MDA8 in three megacity clusters in eastern China, namely BTH, YRD and PRD, showed alarming large upward linear trends of 25%, 10% and 19%, respectively during 2015–2019. Identifying the causes of these worsening $O_3$ trends is urgently required for air pollution prevention and management.

Some recent studies suggested that enhanced photochemical processes induced by changing anthropogenic emissions were

responsible for these trends (Li et al., 2019; Li et al., 2022; Shao et al., 2021; Wang et al., 2020). However, we noticed that there was independent evidence, including the spatial distribution of the expansion of high $O_3$ (Figs. 5 and 6) and inter-annual variations in $O_3$, Ox, $NO_2$, CO and $PM_{2.5}$ (Fig. 8), suggesting that transport/meteorological conditions rather than emissions of $O_3$ precursors were more likely to be the major contributor to the $O_3$ trends. Moreover, we found that the trends in $O_3$ observed in the three megacity clusters during 2015–2020 (Fig. 1) were mainly caused by the large trend of approximately two to three-

fold increase in the number of consecutive $O_3$-exceeding days (Fig. 3), during that time a fast and widespread expansion of high $O_3$ from urban centers to surrounding regions was observed (Fig. 4), and the majority of the expansions was accomplished during the two-year 2015–2017 period (green lines in Fig. 4). Furthermore, the expansions of high $O_3$ in the three megacity clusters were accompanied by a quasi-saturation effect that $O_3$ concentrations at the high $O_3$ stations (high $O_3$ in 2015) of approximate 100 ppb remained nearly constant throughout the entire period of 2015–2020, while the low $O_3$ stations (low $O_3$

in 2015) with $O_3$ less than 75 ppb in all three megacity clusters experienced a significant enhancement in $O_3$ (>5 ppb $yr^{-1}$) during 2015–2017 (Figs. 4a, 4b and 4c). Finally, greater frequency of episodes with four or more consecutive $O_3$-exceeding days contributed the majority to the higher $O_3$ in all three megacity clusters in 2017 vs. 2015, the greater intensity/concentration of $O_3$ during the episodes contributes only about 10% (Fig. 9), consistent with the expansion and quasi-saturation effect discussed earlier.

Coupling the higher values of SSR and T2m in the $O_3$ episodes with four or more consecutive $O_3$-exceeding days depicted in Fig. 10 and greater occurrence (number of days) in the $O_3$ episodes with four or more consecutive $O_3$-exceeding days shown in Fig. 3, we hypothesize that the cause of the worsening $O_3$ trends in BTH, YRD and PRD from 2015 to 2020 could be





attributed to enhanced photochemical $O_3$ production due to the increased occurrence of meteorological conditions of high solar radiation and positive temperature anomaly under the influence of WPSH and TCs. The hypothesis is substantiated in Fig. 11,

which shows excellent correlation between the yearly $O_3$-exceeding days and SSR with R values 0.88 or greater in all three regions. Correlation between the yearly $O_3$-exceeding days and T2m is good in BTH but poor in YRD and PRD, which probably suggests that T2m is not as strongly coupled to $O_3$ formation as SSR.

In conjunction with our study, Hu W. et al. (2023) conducted a statistical analysis to evaluate the processes that promote high $O_3$ formation in PRD when TCs are present in the northwest Pacific. They assessed the impact of the distance between a TC

in the northwest Pacific and PRD on $O_3$ in the PRD from 2006 to 2020. They found that the increased frequency of the downdrafts and stable atmosphere conditions brought forth by the mid-distance category TCs could be the main cause of the large number of consecutive $O_3$-exceeding days in 2019, which contribute to about 80% of the overall positive trend of $O_3$ in 2015–2020 (Fig. 3c and black line, Fig.4c).

Following the analysis of Mao et al. (2020), the impact of WPSH on $O_3$ in BTH in April–September has been analyzed in Fig.

12. We found that the increased frequency of these meteorological conditions enabled by the changing intensity and position of WPSH could contribute as a major cause of the positive $O_3$ trends in the three megacity clusters in eastern China during 2015–2020.

Nevertheless, it is crucial to recognize that the examinations carried out in Sections 3.1–3.3 primarily utilized correlation or regression analysis techniques, which do not inherently establish causal relationships. To attribute cause and effect between

the suggested alterations in meteorological parameters and $O_3$ trends, it is necessary to employ a mechanistic model that accurately replicates the observed $O_3$ trend based on the proposed cause(s). Until the model successfully reproduces the phenomenon, all correlation or regression findings should be treated as merely indicating the highest potential cause-and-effect relationship (Wu et al., 2022).

In conclusion, we hypothesize that the cause of the worsening $O_3$ trends in BTH, YRD and PRD from 2015 to 2020 is

attributable to enhanced photochemical $O_3$ production due to the increased occurrence of meteorological conditions of high solar radiation and positive temperature anomaly under the influence of WPSH and TCs. Therefore, we suggest that future O3 pollution prevention and control policies should pay more attention to changes in the meteorological/climate conditions, particularly changes in the large-scale circulations, including WPSH and TCs.

*Data availability.* Hourly surface $O_3$, $NO_2$, CO, $PM_{2.5}$, and $SO_2$ data were obtained from China National Environmental Centre (http://www.cnemc.cn/en/). Hourly meteorological data are obtained from European Centre for Medium-Range Weather Forecasts ERA5 reanalysis (https://cds.climate.copernicus.eu/). Daily meteorological data are obtained from National Center for Environmental Prediction (NCEP) and National Center for Atmospheric Research (NCAR) (https://psl.noaa.gov/data/gridded/data.ncep.reanalysis.html). The data of this study are available upon request to Shaw Chen

Liu (shawliu@jnu.edu.cn).



*Author Contributions.* SL and RL proposed the essential research idea. TH, and YL performed the analysis. TH, YL, RL, and SL drafted the manuscript. YX, BW, and YZ helped analysis and offered valuable comments. All authors have read and agreed to the published version of the manuscript.


*Competing interests.* The authors declare that they have no conflict of interest.

*Acknowledgments.* The authors thank the China National Environmental Centre and European Centre for Medium-Range Weather Forecasts for providing datasets that made this work possible. We also acknowledge the support of the Institute for
Environmental and Climate Research and Guangdong-Hongkong-Macau Joint Laboratory of Collaborative Innovation for Environmental Quality in Jinan University.

*Financial support.* This research was supported by the National Natural Science Foundation of China (grant number 92044302, 41805115), Guangzhou Municipal Science and Technology Project, China (grant number 202002020065), Special Fund
Project for Science and Technology Innovation Strategy of Guangdong Province (grant number 2019B121205004), Guangdong Innovative and Entrepreneurial Research Team Program (grant number 2016ZT06N263), and National Key Research and Development Program of China (grant number 2018YFC0213906).



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





**Table 1.** Criteria and corresponding numbers of low $O_3$ and high $O_3$ stations in the three megacity clusters in 2015. The criterion listed for each megacity cluster was based on the number of MDA8 $O_3$ exceeding days in 2015. For instance, the criterion for a low $O_3$ site in BTH was the number of MDA8 $O_3$ exceeding days in 2015 being less than or equal to 19 days, while for high $O_3$ site was the number of MDA8 $O_3$ exceeding days in 2015 being greater than or equal to 71 days.

|  | Criterion of low $O_3$ stations | Number of low $O_3$ stations | Criterion of high $O_3$ stations | Number of high $O_3$ stations | Total number of stations |
|---|---|---|---|---|---|
| BTH | ≤19 days | 13 | ≥ 71 days | 14 | 78 |
| YRD | ≤ 37 days | 54 | ≥ 67 days | 13 | 152 |
| PRD | ≤ 12days | 10 | ≥46 days | 10 | 48 |






**Table 2.** Mean $O_3$ concentrations (ppb) and number of days of all $O_3$-exceeding days (2nd column), consecutive $O_3$-exceeding days with less than four days (3rd column), consecutive $O_3$-exceeding days with four or more days (4th column) and the difference between (≥4days) and (<4days) (5th column) within the BTH box in 2015–2020.

|  | All days<br>Concentration (days)<br>ppb | <4 days<br>Concentration (days)<br>ppb | ≥4 days<br>Concentration (days)<br>ppb | Difference<br>(≥4 days) – (<4 days)<br>ppb |
|---|---|---|---|---|
| 2015 | 66.42(31) | 65.04(24) | 71.14(07) | 6.10 |
| 2016 | 64.13(43) | 62.65(26) | 66.39(17) | 3.74 |
| 2017 | 69.44(62) | 65.32(34) | 74.43(28) | 9.11 |
| 2018 | 68.21(74) | 65.43(27) | 69.80(47) | 4.37 |
| 2019 | 70.19(96) | 65.28(30) | 72.42(66) | 7.14 |
| 2020 | 69.69(78) | 65.52(40) | 74.08(38) | 8.56 |





**Table 3.** Same as Table 2, but for YRD.

|  | All days | <4 days | ≥4 days | Difference |
|---|---|---|---|---|
|  | Concentration (days) ppb | Concentration (days) ppb | Concentration (days) ppb | (≥4 days) – (<4 days) ppb |
| 2015 | 53.79(31) | 53.59(19) | 54.11(12) | 0.52 |
| 2016 | 58.87(27) | 58.03(23) | 63.73(04) | 5.70 |
| 2017 | 64.35(40) | 62.62(25) | 67.22(15) | 4.60 |
| 2018 | 63.33(43) | 62.49(32) | 65.75(11) | 3.26 |
| 2019 | 67.18(49) | 66.09(27) | 68.51(22) | 2.42 |
| 2020 | 65.84(38) | 64.12(27) | 70.06(11) | 5.94 |



**Table 4.** Same as Table 2, but for PRD.

|  | All days | <4 days | ≥4 days | Difference |
|---|---|---|---|---|
|  | Concentration (days) ppb | Concentration (days) ppb | Concentration (days) ppb | (≥4 days) – (<4 days) ppb |
| 2015 | 61.16(14) | 61.16(14) | ---(0) | --- |
| 2016 | 58.44(19) | 58.44(19) | ---(0) | --- |
| 2017 | 65.18(36) | 64.60(23) | 66.20(13) | 1.60 |
| 2018 | 65.82(31) | 63.27(16) | 68.55(15) | 5.28 |
| 2019 | 69.80(62) | 65.96(29) | 73.16(33) | 7.20 |
| 2020 | 65.08(37) | 63.87(22) | 66.84(15) | 2.97 |






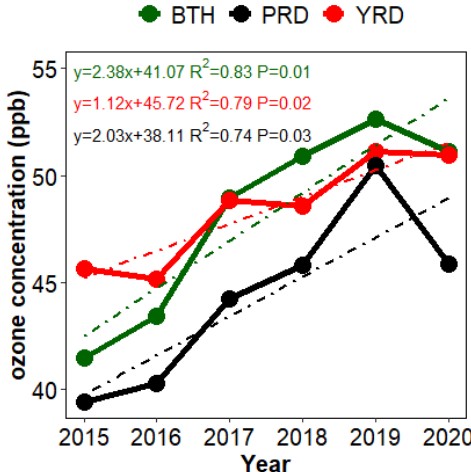

**Figure 1: Annual mean concentrations of maximum daily 8-hour average O₃ in BTH (green), YRD (red) and PRD (black).**



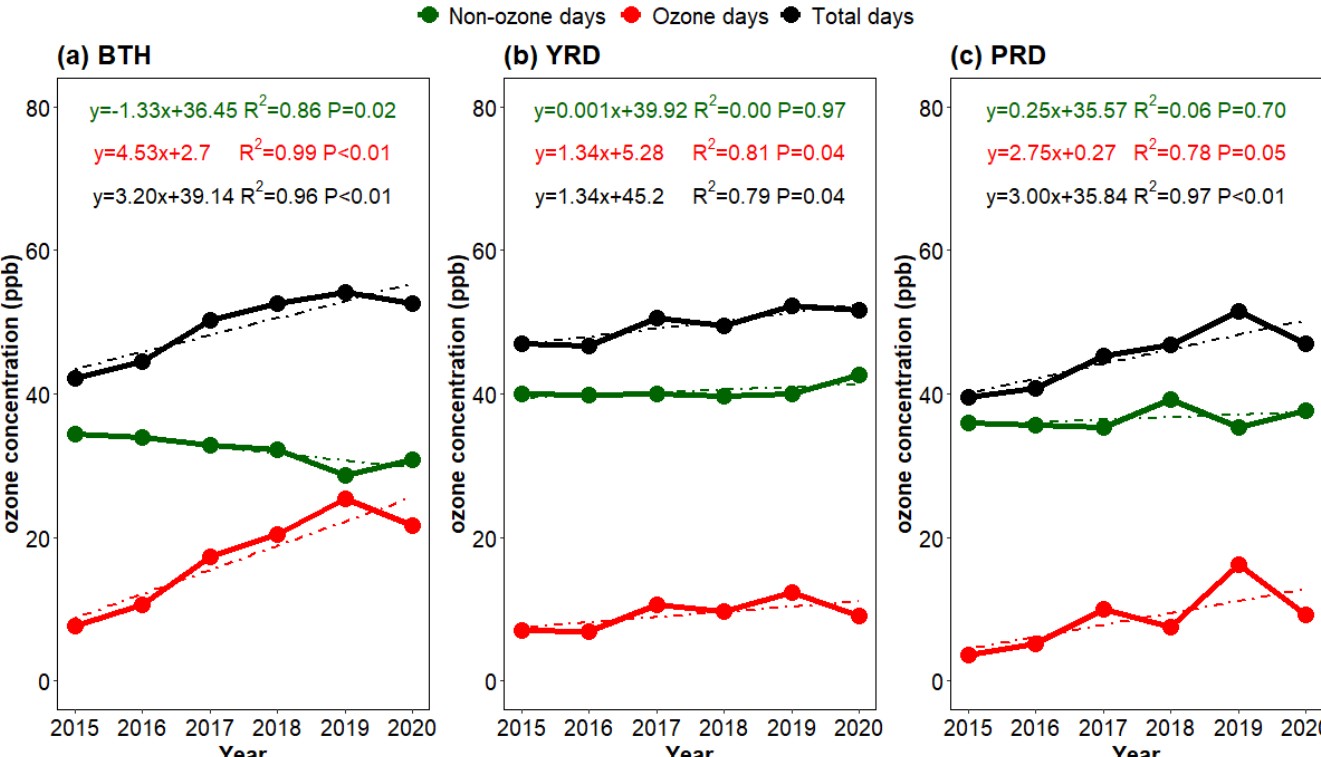

**Figure 2: Contributions from the O₃-exceeding days (red) and non-O₃-exceeding days (green) to the annual mean concentration of maximum daily 8-hour average O₃ (black) in BTH (a), YRD (b) and PRD (c).**



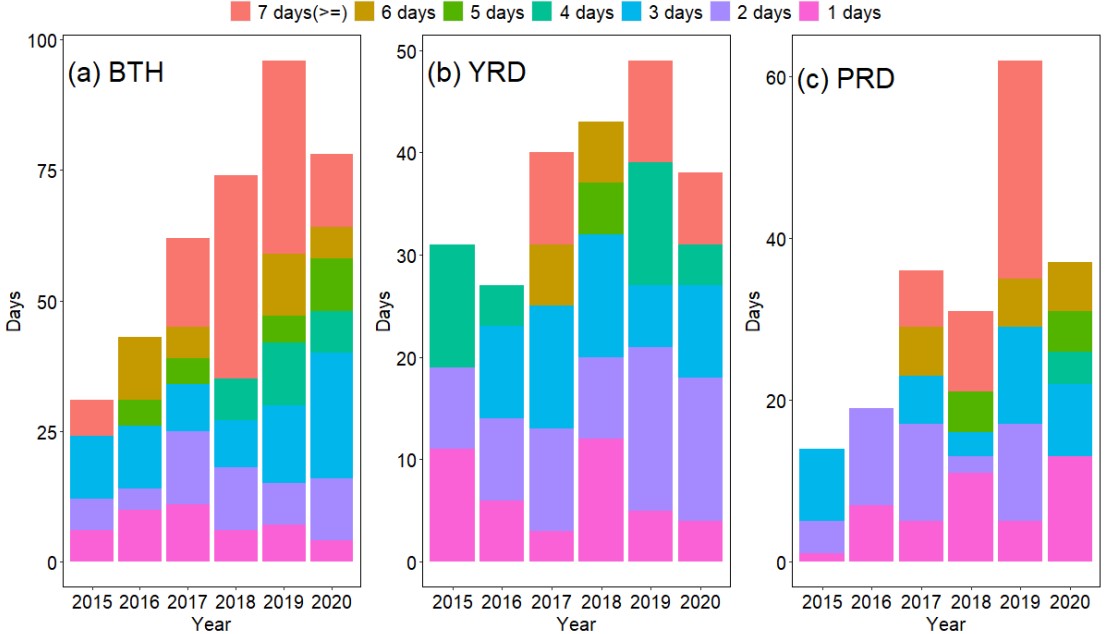

**Figure 3: Annual numbers of various consecutive O$_3$-exceeding days in BTH (a), YRD (b) and PRD (c). Individual colors denote different numbers of consecutive O$_3$-exceeding days.**



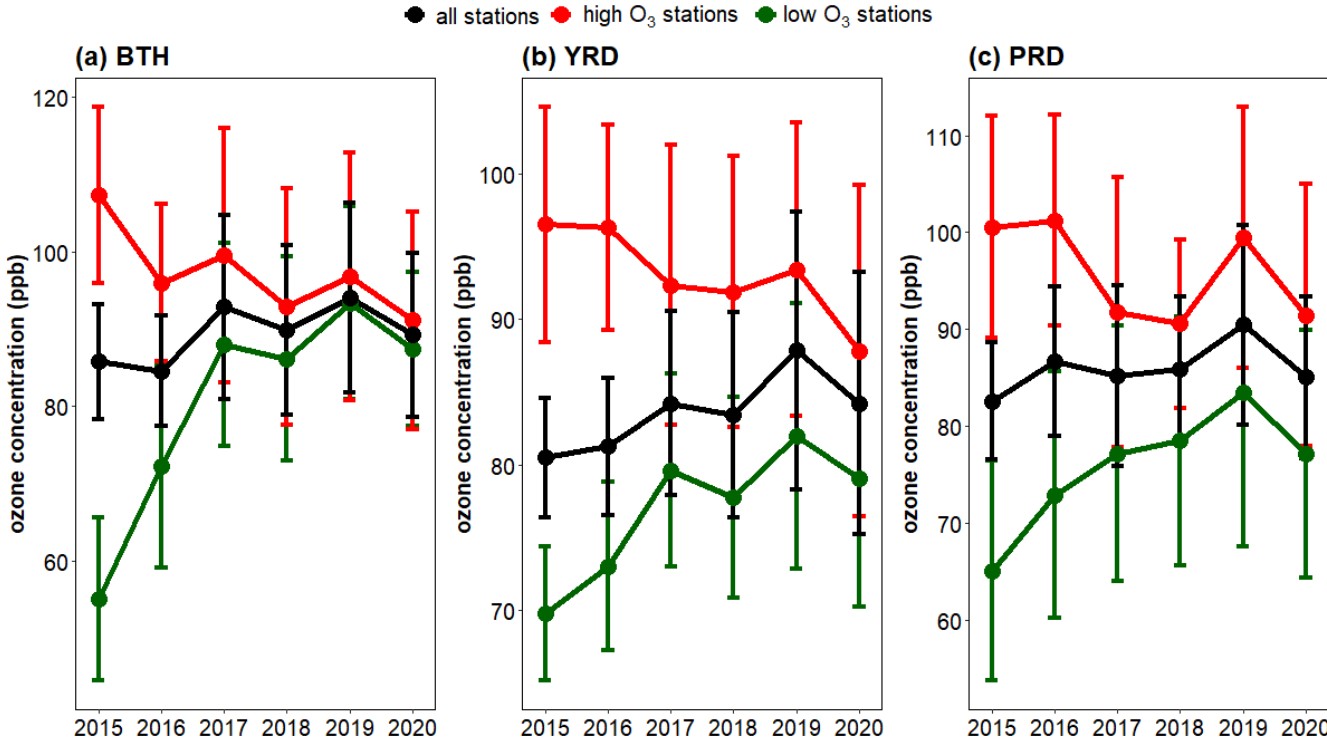

**Figure 4: Mean concentrations of maximum daily 8-hour average O₃ during O₃-exceeding days for all stations (black), high O₃ stations (red) and low O₃ stations (green) in BTH (a), YRD (b) and PRD (c).**



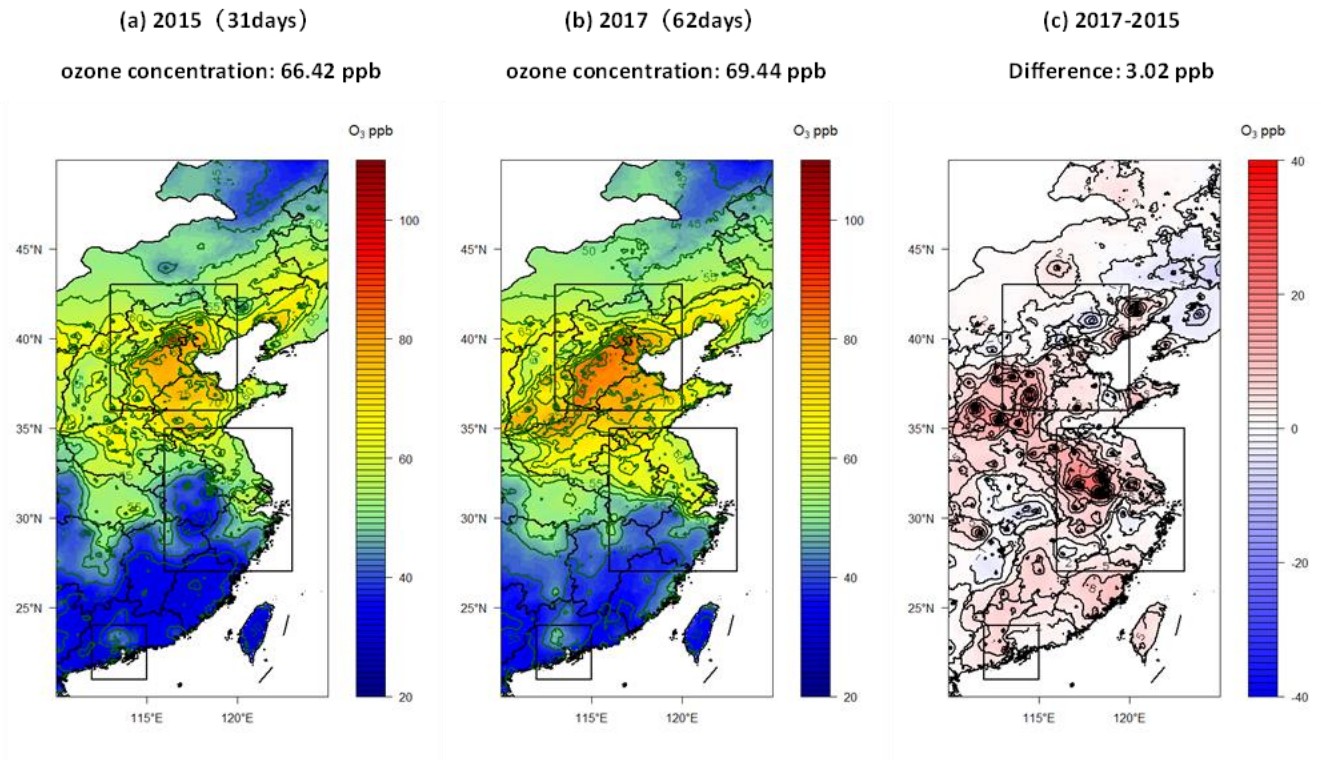

**Figure 5: Spatial distribution of annual mean concentrations of maximum daily 8-hour average O₃ for O₃-exceeding days in BTH in 2015 (a), 2017 (b) and their difference (2017 - 2015) (c). The top, middle and bottom rectangle boxes denote BTH, YRD and PRD districts, respectively. The number inside the parenthesis behind 2015 or 2017 denotes the number of O₃-exceeding days.**



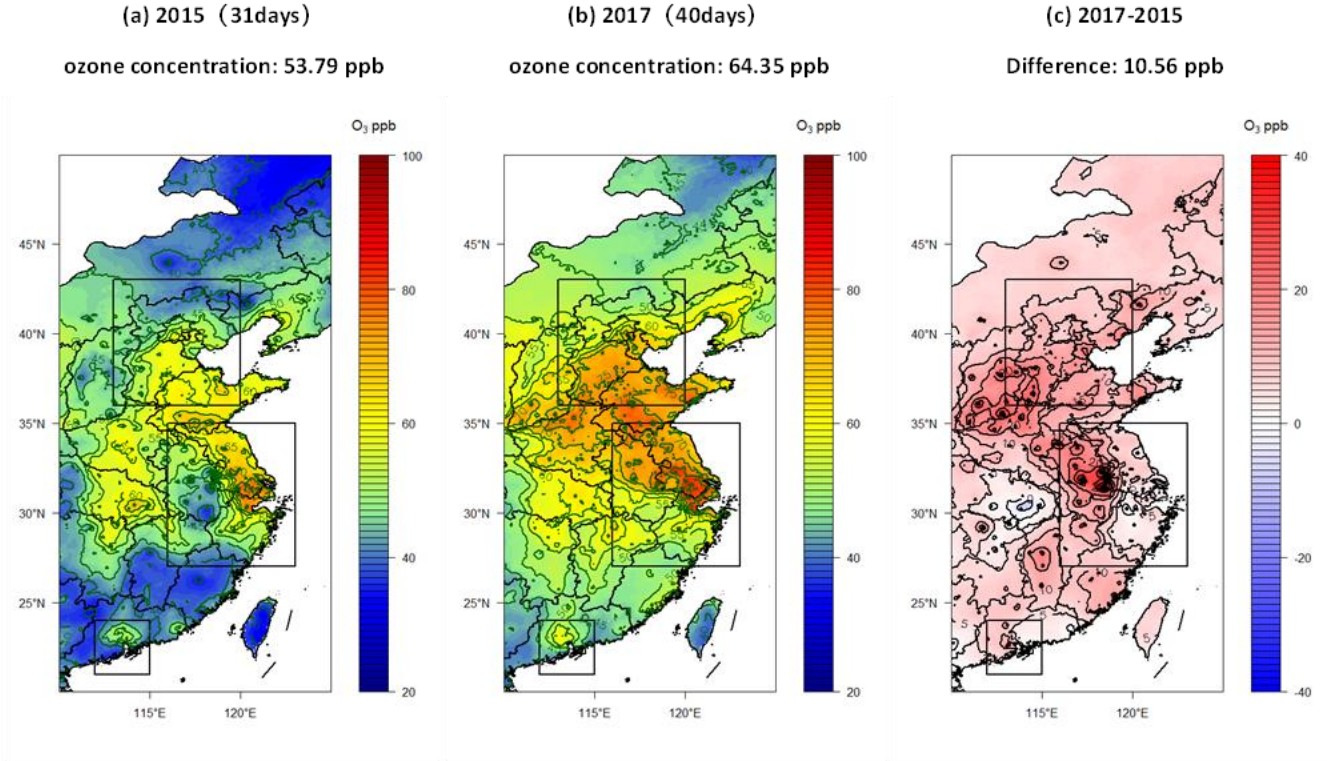

**Figure 6: Same as Figure 5 except for YRD.**




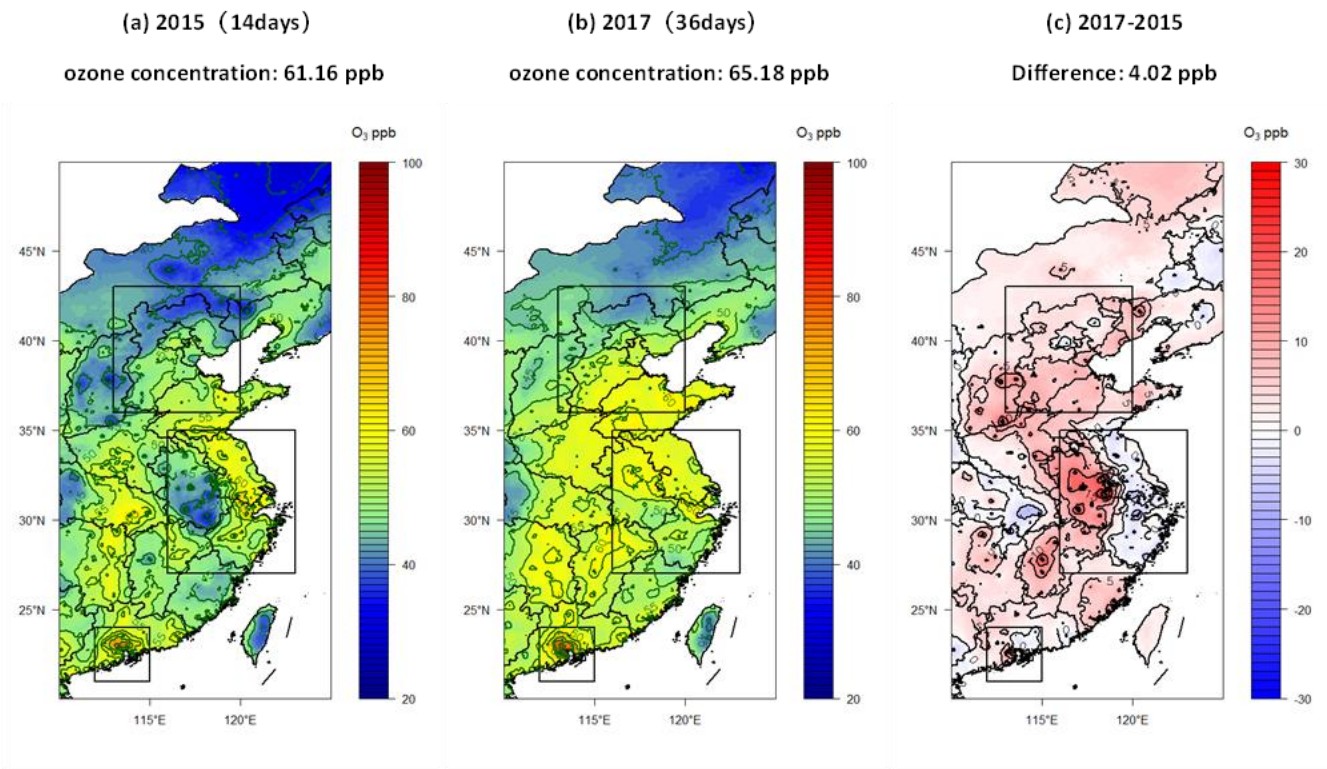

**Figure 7: Same as Figure 5 except for PRD.**



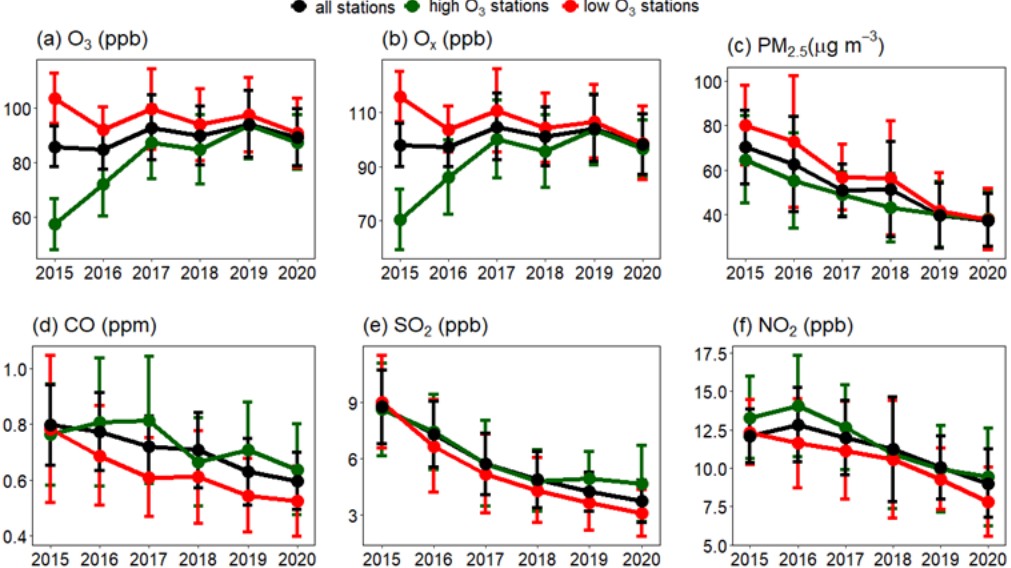

**Figure 8: Annual mean concentrations of maximum daily 8-hour average O₃ in BTH during O₃-exceeding days for all stations (black), high O₃ stations (red) and low O₃ stations (green) (a), same as (a) except for Ox (b), PM₂.₅ (c), CO (d), SO₂ (e), NO₂ (f).**





**Figure 9: Spatial distribution of daily mean MDA8 O₃ of O₃-exceeding days in BTH for O₃ episodes with four or more consecutive O₃-exceeding days in 2015 (a), O₃ episodes with less than four consecutive O₃-exceeding days in 2015 (b), and (a minus b) (c); (d, e and f) are the same as (a, b and c), respectively, except for 2017.**





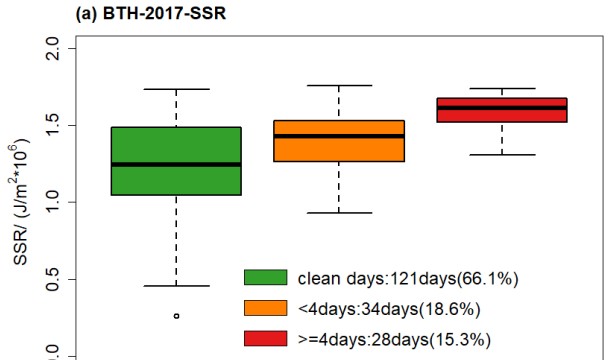
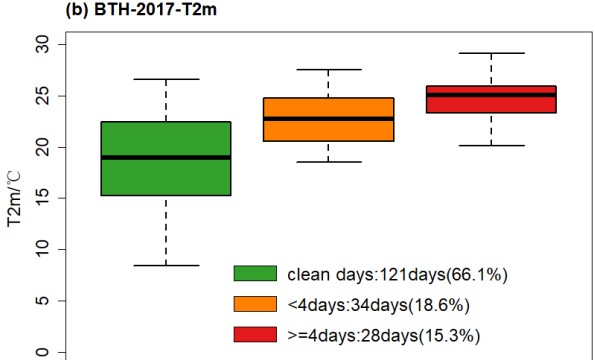

**Figure 10: Surface solar radiation (SSR) (a) and temperature (T2m) (b) in BTH in April–September 2017 for four episodes with four or more consecutive O$_3$-exceeding days (red), clean days (non-O$_3$-exceeding days) (green) and O$_3$ episodes with less than four consecutive O$_3$-exceeding days (orange).**




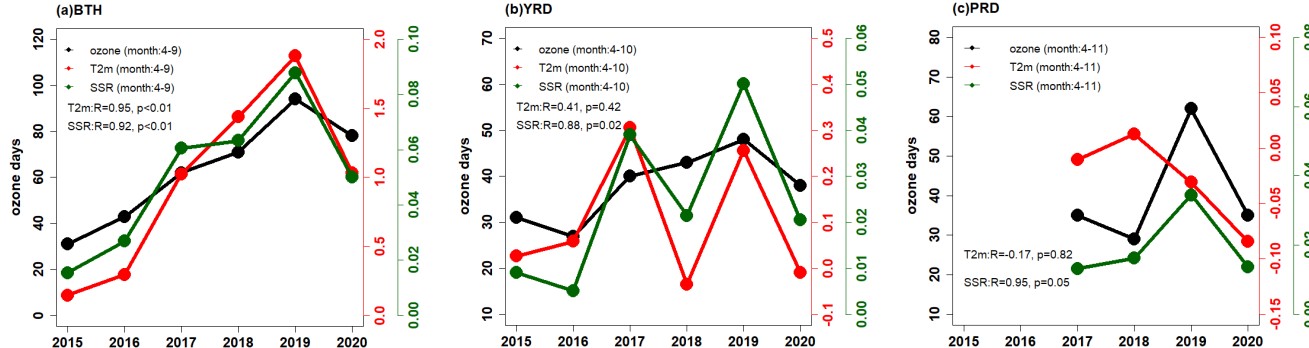

**Figure 11. Correlations among annual O₃-exceeding days, surface solar radiation (SSR) and temperature (T2m) in BTH (a), YRD (b) and PRD (c).**



**(a) 2015:all O$_3$ days**

**(86ppb, 31 days)**

**(b) 2015:clean days**

**(52ppb, 152 days)**

**(c) 2017:all O$_3$ days**

**(93ppb, 62 days)**

**(d) 2017:clean days**

**(56ppb, 121 days)**

**(e) 2019:all O$_3$ days**

**(94ppb, 95 days)**

**(f) 2019:clean days**

**(54ppb, 89 days)**

**Figure 12: Composite 500 hPa geopotential height contours, humidity and winds in BTH in April-September for O$_3$-exceeding days in 2015 (a), clean days in 2015 (b), O$_3$-exceeding days in 2017 (c), clean days in 2017 (d), O$_3$-exceeding days in 2019 (e) and clean days in 2019 (f).**




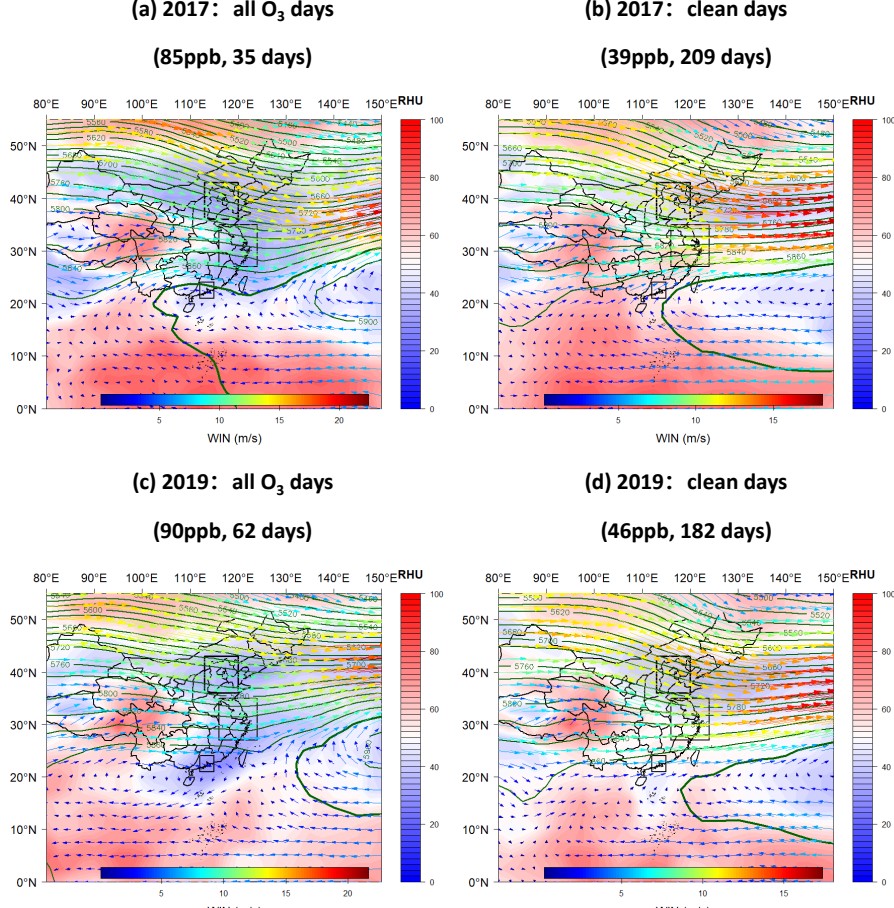

**Figure 13: Composite 500 hPa geopotential height contours, humidity and winds in PRD in April-November for O₃-exceeding days in 2017 (a), clean days in 2017 (b), O₃-exceeding days in 2019 (c), clean days in 2019 (d).**