# Peer review of "What is the cause(s) of positive ozone trends in three megacity clusters in eastern China during 2015–2020?"

_EGUsphere, 2023_

## Author Comment (AC1)

Dear Editor,

We appreciate the prompt reviews and would like to thank the reviewers for insightful comments and suggestions on our manuscript entitled "What is the cause(s) of positive ozone trends in three megacity clusters in eastern China during 2015–2020?" (MS No.: egusphere-2023-1088). We have carefully considered all comments and suggestions. Listed below are our point-by-point responses to all comments and suggestions of Reviewer #1 (Reviewer's points in black, our responses in blue). Extensive revisions have been made in the revised manuscript to address the comments and suggestions, which we believe is significantly improved. For that we thank the reviewer.

**Anonymous Referee #1**

This paper analyzes the cause of 2015-2020 positive ozone trends in megacities in China. While the topic is important, the main conclusion of this study is not well supported by the analyses. In addition, the presentation, including the logic, word expression, and figure quality requires substantial improvement. I recommend at least a major revision before it can be re-considered to be published in ACP.

**Response:**

In the followings, we have carefully considered and responded to your comments and suggestions.

**Main concern:**

The authors emphasize the role of meteorology and diminish that of emission change as the key cause of the ozone rise between 2015-2020 in many places of the text, but in most places, there is no direct evidence to approve or disapprove, and the conclusions seem to be arbitrary, I list several questions related to this point below that need to be addressed.

(1) Line 166-167: Why is this simply attributed to weather system? (e.g. lines 166-168,

178-180, and many others).

**Response:**

Thank you for pointing out the lack of clarity on attributing the cause of interannual variability of $O_3$ in our paper. We clarify this problem in the revised manuscript by consolidating and elaborating statements in "lines 166–168, 178–180, and many others" into a new section near line 243 as shown below.

**"3.3.1 Changing emissions as a possible cause of $O_3$ trends in 2015–2020**

As mentioned earlier, two emission-oriented hypotheses have been proposed as a possible cause of the $O_3$ trends in 2015–2020. One is changing emissions of $O_3$ precursors NOx and VOC (Li et al., 2022). The other is the reduced removal of $HO_2$ radicals by diminishing $PM_{2.5}$ suggested by Li K. et al. (2021) and Shao et al. (2021). Li et al. (2022) demonstrated convincingly that the NO titration effect was the cause of the linear trend in $O_3$ in PRD (0.5 ppb $yr^{-1}$) during the period 2006–2019. But for the period 2015–2020, the NO titration effect could account for only about 10% of the linear trend in $O_3$ of the low $O_3$ stations in PRD (5.0 ppb $yr^{-1}$, green line, Fig. S3a).

The increase of 30 ppb in $O_3$ at the low $O_3$ stations in BTH from 2015 to 2017 (green line, Figs. 4a and 8a represents about 50% increase in $O_3$. The titration effect can account for only about 5% (Fig. 8f). If this increase of 30 ppb in $O_3$ were due to an enhancement in $O_3$ precursors, the enhancement would have to be substantially greater than 50% because of the well-known less-than-linear relationship between changes in $O_3$ and its precursors, i.e., substantially more percentage changes in precursors are needed for each percentage change in $O_3$ (Dodge, 1977; Shafer and Seinfeld, 1985). Figures 8d and 8f show that CO (a proxy for VOC) and NOx changed only by a few percent from 2015 to 2017, more than one order of magnitude less than the changes needed. Hence it appears that changes in meteorological conditions conducive to $O_3$ formation are more likely the major contributing factor to the 50% increase in $O_3$ at the low $O_3$ stations in BTH. Similar argument can be extended to YRD and PRD (Figs. S1

and S3).

The theory of reduced removal of $HO_2$ radicals by diminishing $PM_{2.5}$ (25%, green line of Fig. 8c) appeared to be valid qualitatively for the 50% increase in $O_3$ at the low $O_3$ stations in BTH from 2015 to 2017 (green line of Fig. 8a). But this theory was contradicted directly by the phenomenon at the high $O_3$ stations where a 30% reduction in $PM_{2.5}$ (red line of Fig. 8c) corresponded to a decrease rather than an increase in $O_3$ (red line of Fig. 8a)."

[Figure]

**Figure S1.** Annual mean concentrations of maximum daily 8-hour average $O_3$ in YRD during $O_3$-exceeding days for all stations (black), high $O_3$ stations (red) and low $O_3$ stations (green) (a), same as (a) except for Ox (b), $PM_{2.5}$ (c), CO (d), $SO_2$ (e), $NO_2$ (f).

[Figure]

**Figure S3.** Annual mean concentrations of maximum daily 8-hour average $O_3$ in PRD during $O_3$-exceeding days for all stations (black), high $O_3$ stations (red) and low $O_3$ stations (green) (a), same as (a) except for Ox (b), $PM_{2.5}$ (c), CO (d), $SO_2$ (e), $NO_2$ (f).

(2) Line 177-180: For what reason it is "highly unlikely emission change"? It is not convincing to guess the cause simply based on the spatial pattern of ozone change. In addition, the ozone data used in this study is smoothed from observations in the TAP dataset, thus it may not reflect the true ozone change in the regions with no direct observation.

**Response:**

This part of the text (Line 177–180) has been rewritten entirely in the revised manuscript. Please see our response to your Main concern (1) about "highly unlikely emission changes".

The TAP data system integrates multiple data sources, including ground observations, satellite remote sensing data, high-resolution emission inventories, air quality model simulations, and other relevant information (Xue et al., 2020; Xiao et al., 2022). Over time, the TAP team has made consistent enhancements, resulting in a three-stage $O_3$

prediction model with an impressive $R^2$ value of 0.84 when compared to ground observations spanning the years 2013 to 2020. Consequently, we have confidence that the spatial $O_3$ data provided by TAP effectively captures the variations in $O_3$ concentrations within the megacity clusters.

(3) The function of Section 3.2 is very confusing to me. The paper is about 2015-2020 trends, but here only the difference between 2015 and 2017 is analyzed. Even though the ozone difference between 2017 and 2015 is mostly driven by weather anomalies, the authors do not explain what weather system can sustain high ozone from 2017 to 2018-2020. Analyses of yearly differences cannot be simply applied to explain the 6-year trend.

**Response:**

We understand your concern regarding the focus on the difference between 2015 and 2017. The time period we are examining spans only six years (2015–2020), a duration that might not typically be referred to as a trend if not for its significant environmental impact. The two years between 2015 and 2017 represent a substantial portion of the entire period, contributing one third or more to the overall trends in the three megacity clusters (as depicted in Fig. 4, black lines). In fact, if the $O_3$ levels in 2017 and 2019 had been the same as those in 2015, there would have been no discernible trend in any of the three megacity clusters (Fig. 4, black lines). Therefore, to understand the six-year trend, it is crucial to elucidate the factors behind the elevated $O_3$ levels in 2017 and 2019. Furthermore, it's worth noting that only the high values of 2019 are essential for the PRD region (Fig. 4c, black lines). We now have a good understanding that these elevated levels were influenced by the high frequency of downdrafts and stable atmospheric conditions associated with tropical cyclones in the northwest Pacific (as detailed in Section 3.3.2 of the revised manuscript).

In addition, in Section 3.3.3 we have conducted a comprehensive analysis of the role played by the Western Pacific Subtropical High (WPSH) during $O_3$-exceeding days in

all three megacity clusters for each year from 2015 to 2020.

(4) The authors list "ozone at high ozone stations unchanged while ozone at low ozone station increase" as a major finding (Line 235). If it is driven by weather, I wonder what weather system could selectively influence sites with different ozone levels. It more likely reflects chemical factors. This is a key concern.

**Response:**

We appreciate this "key concern" as it points out a lack of clarity in our presentation. We actually fully agree that "It more likely reflects chemical factors." But we believe that the "chemical factors" are driven by changes in meteorology rather than changes in emissions of air pollutants. As a photochemical product, the ozone trends are primarily controlled by the chemical production of ozone. The contribution of meteorology proposed in our manuscript is mostly through enhanced photochemical production of ozone at the low $O_3$ stations. This point was stated at lines 23–26 in the abstract which was made in response to a previous similar concern of the editor-in-charge. In addition, we have revised lines 235–243 as follows: "And (3), the expansions of high $O_3$ in the three megacity clusters were accompanied by a saturation effect that $O_3$ concentrations at the high $O_3$ stations of approximate 100 ppb in 2015 remained nearly constant or slightly declined throughout the entire period of 2015–2020 (Fig. 4)."

In regard to "what weather system could selectively influence sites with different ozone levels"? We address this point by adding the following four paragraphs and two figures (Figs. 12 and 13) near Line 300:

[revised manuscript text omitted]

(5) Line 210-223: VOCs emission change is not considered here. And again this is for difference between 2015 and 2017, it doesn't explain the trend at all.

**Response:**

Please refer to our response to Main concern (3). Specifically, Lines 210–223 has been

replaced in the revised manuscript by Lines 216–228 in the new section 3.3.1, which does consider VOC emissions as shown below: "The increase of 30 ppb in $O_3$ at the low $O_3$ stations in BTH from 2015 to 2017 (green line, Figs. 4a and 8a) represents about 50% increase in $O_3$. The titration effect can account for only about 5% (Fig. 8f). If this increase of 30 ppb in $O_3$ were due to an enhancement in $O_3$ precursors, the enhancement would have to be substantially greater than 50% because of the well-known less-than-linear relationship between changes in $O_3$ and its precursors, i.e., substantially more percentage changes in precursors are needed for each percentage change in $O_3$ (Dodge, 1977; Shafer and Seinfeld, 1985). Figs. 8d and 8f show that CO (a proxy for VOC) and NOx changed only by a few percent from 2015 to 2017, more than one order of magnitude less than the changes needed. Hence it appears that changes in meteorological conditions conducive to $O_3$ formation are more likely the major contributing factor to the 50% increase in $O_3$ at the low $O_3$ stations in BTH. Similar argument can be extended to YRD and PRD (Figs. S1 and S3).

The theory of reduced removal of $HO_2$ radicals by diminishing $PM_{2.5}$ (25%, green line of Fig. 8c) appeared to be valid qualitatively for the 50% increase in $O_3$ at the low $O_3$ stations in BTH from 2015 to 2017 (green line of Fig. 8a). But this theory was contradicted directly by the phenomenon at the high $O_3$ stations where a 30% reduction in $PM_{2.5}$ (red line of Fig. 8c) corresponded to a decrease rather than an increase in $O_3$ (red line of Fig. 8a)."

(6) Section 3.3 is not convincing as well. It should try to explain what weather system contributes to more ozone consecutive day from 2015 (as authors state that it drives the ozone increase), but figures are not helpful for this purpose. Figure 11 only shows that meteorological parameters can explain some of the ozone variability, Figures 12-13 show that WPSH can influence weather patterns, but is there any hint of an increasing influence of WPSH on consecutive ozone days? It might be useful to first clarify the weather patterns for consecutive ozone days, and explain what system can explain an increase in the frequency of such weather pattern during 2015-2020.

**Response:**

We take this comment to heart and have made an extensive revision of Section 3.3 as follows. First, as shown in our response to your major question (1), a new section 3.3.1 on "Changing emissions as a possible cause of $O_3$ trends in 2015–2020" has been added. Second, in our response to your "key concern" (4), we have added 39 lines (~750 words) of discussion and two figures (Figs. 12 and 13) near Line 300 "to first clarify the weather patterns for consecutive ozone days and explain what weather system contributes to more ozone consecutive day from 2015". Finally, we have added the following two paragraphs to Section 4 Summary and conclusions: "The trends in $O_3$ in the three megacity clusters are found to be critically dependent on the number of four or more consecutive $O_3$-exceeding days. In collaboration with this study, Hu W. et al. (2023) found that the changing frequency of mid-distance category TCs (i.e. changing meteorological conditions) is the cause of the increases in the numbers of consecutive $O_3$-exceeding days as well as the $O_3$ concentrations in PRD. Our additional analyses of the mean vertical velocity at 850 hPa in the three megacity clusters (Figs. 12, 13 and S10) show that the increases in $O_3$ in all three megacity clusters from 2015 to 2017 were associated with enhanced downdrafts and stable atmospheric conditions (meteorological conditions) which were highly conducive to $O_3$ formation. Finally, the enhanced downdrafts and stable atmospheric conditions were most likely brought about by TCs and associated WPSH.

Therefore, we propose that the $O_3$ concentrations at the high $O_3$ stations stayed close to a saturation level of about 100 ppb throughout 2015 to 2020, even under enhanced conditions conducive to $O_3$ formation, was the result of a relatively high rates of atmospheric dispersion, dry deposition and photochemical loss at the high $O_3$ concentration. While the low $O_3$ stations, where $O_3$ production were relatively small in 2015, experienced significant enhancements in the $O_3$ production in 2017 and 2019 because of the enhanced downdrafts and stable atmospheric conditions associated with TCs and WPSH in the northwestern Pacific, which were highly conducive to $O_3$ formation (Hu W. et al., 2023).".

We acknowledge that this study, like most other investigations, is an ongoing research effort. We report here some new results, but we don't have the definitive answers to all mechanisms that lead to the increasing "weather patterns for consecutive $O_3$ days" in the three megacity clusters. More studies are needed to address those questions.

**Other comments**

Line 100, Before Table 1, please consider introducing the purpose for such classification.

**Response:**

Thanks. we have revised the manuscript as follows to address the purpose of classification:

"Table 1 lists the criteria and corresponding numbers of low $O_3$ and high $O_3$ stations in the three megacity clusters. This classification is undertaken with the purpose of distinguishing stations with various $O_3$ levels within the three megacity clusters, and it is based on the number of $O_3$ exceeding days in 2015."

Line 116: missing ppb

**Response:**

Thanks. We have added "ppb".

Line 158-160. It is quite unclear what the calculation stands for, and how it leads to the conclusion that. Please clarify. Please also carefully clarify the rationale of other formulas.

**Response:**

In section 3.2, if we only compare the MDA8 $O_3$ average concentration on $O_3$ exceedance days between two years, the difference between 2017 and 2015 is only 3.02 ppb. However, considering that in 2015 there were only 31 days with $O_3$ exceedances,

while in 2017 there were 62 days, characterizing the severity of $O_3$ exceedances between the two years solely based on the MDA8 $O_3$ average concentration on exceedance days is clearly insufficient. Therefore, we also take into account of the number of $O_3$ exceedance days and calculate a normalized MAD8 $O_3$ concentration by multiplying the MDA8 $O_3$ on exceedance days by the number of exceedance days and dividing by the total number of days in each year. When we compare the normalized MAD8 $O_3$ concentration for 2017 and 2015 (Fig. 2, red lines), we obtain a ratio of 2.09 between the two years.

To make it clearer, we have revised the corresponding statements as follows:

"The daily average concentration of MDA8 $O_3$ within the BTH box increased from 66.42 ppb in 2015 (31 days, Fig. 5a) to 69.44 ppb in 2017 (62 days, Fig. 5b), which was a difference of 3.02 ppb or a merely 4.5% increase between the two years (Fig. 5c). After accounting for the number of $O_3$-exceeding days, the ratio of normalized MDA8 $O_3$ in all $O_3$-exceeding days between 2017 and 2015 became 2.09. This comparison suggests together with those shown in Fig. 3a suggest that the increase in $O_3$ in BTH between 2015 and 2017 was driven primarily by the increase of consecutive $O_3$-exceeding days."

The statements pertaining to YRD and PRD have been adjusted accordingly.

Figure 3: Should "episodes" be a better word compared to "days" for the y-axis, since the variables plotted are "days"?

**Response:**

Thanks. We have added "Episode" to the title.

The expression needs to be improved, and the use of words needs re-consideration. For example, what does "quasi-saturation" stand for? Please revise "Same as Figure 5 except for YRD" to "Same as Figure 5 but for YRD". Please carefully check others.

**Response:**

Thanks. We have replaced some of the expressions, and revised "Same as Figure 5 except for YRD" to "Same as Figure 5 but for YRD" and others.

In this study, "quasi-saturation" refers to the phenomenon that $O_3$ concentration becomes nearly saturated and stops increasing after reaching certain level (approximate 100 ppb). Since both reviewers question the word "quasi-saturation", we change it to simply "saturation".

Figure quality can be improved, by increasing resolution and avoiding contours on shadings if both represent the same variable (Figs 5-6)

**Response:**

Thanks. We have revised all figures in the revised manuscript.

---

## Author Comment (AC2)

Dear Editor,

We appreciate the prompt reviews and would like to thank the reviewers for insightful comments and suggestions on our manuscript entitled "What is the cause(s) of positive ozone trends in three megacity clusters in eastern China during 2015–2020?" (MS No.: egusphere-2023-1088). We have carefully considered all comments and suggestions. Listed below are our point-by-point responses to all comments and suggestions of Reviewer #2 (Reviewer's points in black, our responses in blue).

**Anonymous Referee #2**

The authors explore the potential mechanism driving the observed increase in surface Ozone during 2015-2020 over three megacity clusters in eastern China. Observational data for several pollutants from the Chinese National Environmental Ministry of Environmental Protection and Tracking Air Pollution in China dataset are analyzed in the paper to explore the trends in surface ozone over the study regions for the period of interest. Further, reanalysis data from ERA5, NCEP and NCAR are used to investigate the correlation between the evolving weather systems and the positive ozone trends. The study approach is mainly based on statistical analysis of observational data.

**General comments:**

The paper presents the meteorological conditions conducive to ozone formation (e.g., increased solar radiation) as potentially driving the positive ozone trends, rather than an increase in the anthropogenic emissions or a combination of both. While this is an important and interesting topic, some concerns need to be addressed in the paper before publication. The paper section structure, wording and logic, and the overall presentation can be further improved. More information needs to be included in the paper to further support the hypothesis that weather systems and changing meteorological conditions are responsible for observed increase in $O_3$. Please consider the following suggestions:

1. List the processes (e.g., photochemistry) and precursors involved in the production

of ozone, explicitly with tables and/or graphs (e.g., EKMA ozone isopleth diagram). Explicitly show the correlation (even if a weak correlation) between emissions of precursors (e.g., NOx, VOCs) and $O_3$ levels.

**Response:**

A highly simplified $O_3$ production scheme can be shown as follows:

$$OH + CO(VOC) + O_2 \rightarrow HO_2(RO_2) + CO_2$$

$$HO_2(RO_2) + NO \rightarrow NO_2 + OH(RO)$$

$$NO_2 + hv(\lambda < 420nm) \rightarrow NO + O(^3P)$$

$$O(^3P) + O_2 + M \rightarrow O_3 + M$$
* * *
$$\text{Net:} \quad CO(VOC) + 2O_2 + hv(\lambda < 420nm) \rightarrow CO_2 + O_3 + (RO)$$

Where $NO_x$ (NO + $NO_2$) and $HO_x$ (OH + $HO_2$) act as catalysts for $O_3$ production. Generally, the $O_3$ production tends to go up with the catalysts. In reality, other reactions can become competitive with the reactions above under high NOx conditions, and thus reduce the $O_3$ production efficiency. For instance, the reaction $OH + NO_2 + M \rightarrow HNO_3 + M$ can reduce $HO_x$ and $O_3$ at high concentrations of $NO_2$. In addition, the titration of $O_3$ by NO becomes effective when NO emissions are high.

To explicitly show the correlation between emissions of precursors and $O_3$ levels, we list the correlation coefficient as follows:

Table R1. Correlation coefficients between precursors and $O_3$ in three megacity clusters.

|  | BTH | YRD | PRD |
| --- | --- | --- | --- |
| NOx | -0.30(p-value =0.56) | -0.82(p-value =0.04) | -0.03(p-value =0.96) |
| VOCs | -0.59(p-value =0.22) | -0.15(p-value =0.78) | -0.28(p-value =0.58) |

[Figure]

**Figure R1.** Annual mean concentrations of maximum daily 8-hour average $O_3$ in YRD during $O_3$-exceeding days for all stations (black), high $O_3$ stations (red) and low $O_3$ stations (green) (a), same as (a) except for Ox (b), $PM_{2.5}$ (c), CO (d), $SO_2$ (e), $NO_2$ (f).

[Figure]

**Figure R2.** Annual mean concentrations of maximum daily 8-hour average $O_3$ in PRD during $O_3$-exceeding days for all stations (black), high $O_3$ stations (red) and low $O_3$

stations (green) (a), same as (a) except for Ox (b), PM$_{2.5}$ (c), CO (d), SO$_2$(e), NO$_2$ (f).

[Figure]

**Figure R3.** NOx and VOCs emissions in the BTH, YRD and PRD regions from 2015 to 2020.

2. Compare the meteorological conditions to longer range time periods to clearly demonstrate that conditions have evolved towards increased ozone production. Comment on why these conditions have changed.

**Response:**

We appreciate this suggestion. A similar comment was made by Reviewer 1, Main concern (4). We address this point by adding the following four paragraphs and two figures (Figs. 12 and 13) near Line 300:

[revised manuscript text omitted]

3.  Include information on how land use/development was changed during the same period, to compare against the spatial expansion of high $O_3$ from urban centers to surrounding regions (past vs current).

**Response:**

We did not discuss changes in land use/development during the study period in our manuscript for the following reasons:

(1) Over a relatively short time (7 years) frame, especially in already highly developed areas like BTH, YRD and PRD, significant changes in land use are not expected. Additionally, acquiring information on short-term land use/development changes is not practical due to lack of relevant data.

(2) In line with China's Technical Regulation for the Selection of Ambient Air Quality Monitoring Stations (on trial) (HJ 664-2013) (Ministry of Environmental Protection, 2013), it is imperative that ambient air quality monitoring stations strategically

incorporate considerations from both urban and rural development plans. This approach ensures that identified monitoring locations account for the evolving spatial patterns in urban and rural areas over time, guaranteeing their representativeness. These stations are tasked with objectively depicting ambient air quality levels and trends within a defined geographical area and providing an accurate assessment of the influence of pollution sources on local air quality. Additionally, these stations placements must factor in various environmental aspects such as physical geography, meteorology, as well as socioeconomic characteristics including industrial distribution and population density. Their purpose is to accurately portray the current state and future trends of air quality within key functional zones and primary sources of air pollution in the city. It's worth noting that from 2015 to 2020, the number of monitoring stations within the study area has remained constant, and the ability of these stations to reflect their surrounding conditions should likewise remain unchanged.

(3) The selection of different monitoring sites was not based on land use/development criteria but rather on the $O_3$ pollution levels at each site in the initial year of this study, which was 2015.

4. Authors acknowledge that their results are mainly based on statistical correlations and further investigation into causal relationships is needed, perhaps with a use of a chemical/transport model. I agree and I'd like to emphasize that this topic is a great case for a model-based investigation, although it might be out of scope for the current manuscript. Model scenario simulations, with different input emissions and for various meteorological conditions, are crucial for further investigating this topic. All models have limitations, but their power and capability in investigating air quality and transport scenarios cannot be dismissed.

**Response:**

Thanks for this insightful comment. Yes, we fully agree that a realistic 3-D model of $O_3$ would be an ideal tool to determine the relative contributions of emission and/or

meteorology to the linear trends of $O_3$ in the three megacity clusters. We have tried many modeling studies on $O_3$ and have come to the recognition that current models have too large uncertainties and limitations to simulate the $O_3$ trends of the three megacity clusters realistically. To our knowledge, no modeling study has reported any successful simulation of the $O_3$ trends in the three megacity clusters.

Specific comments:

1. Line 23: "These favorable meteorological conditions greatly facilitated the formation of $O_3$" - suggests causal relationship, while only correlation is established in the paper…, please consider revising.

**Response:**

Please refer to our response to your general comment #2. In the extensive revisions in response to this general comment, we have linked mechanically the significant increases in the $O_3$ production in 2017 and 2019 to the enhanced downdrafts and stable atmospheric conditions associated with corresponding changes in TCs and WPSH in the northwestern Pacific.

2. Line 37: "The concentrations of air pollutants $SO_2$, NOx, CO, $PM_{10}$ and $PM_{2.5}$ in China have been significantly reduced since 2013" – what about VOCs?

**Response:**

The emissions of VOCs have also declined since 2013 (Fig. R3). Because VOCs are not currently included in China's regular pollutants, their trends were not mentioned in the original manuscript.

3. Figure 1,2,3,4: What caused the reduction in 2020? The Covid19 pandemic related closures and slowed down activities perhaps? You can see the same reduction in Figures 2, 3 and 4. Is this related to decreased emissions of precursors in 2019-2020 or changing meteorological conditions?

**Response:**

In the three regions, the $O_3$ concentrations and $O_3$ exceedance days in 2018 were all lower than those in 2019, but were comparable to those of 2017 and 2020. These features suggest that changing meteorology rather than the 2020 COVID-19 pandemic was more likely responsible for the interannual variations in $O_3$.

In fact, there were many studies reporting that during the early stages of the COVID-19 lockdown in early 2020, $O_3$ concentration did not decrease but instead increased (Huang et al., 2020; Le et al., 2020). We believe that the decline in 2020 compared to 2019 was mainly due to changes in meteorological conditions. The number of rainy days from May to October in the BTH and YRD in 2019 was the lowest in 2015, and the radiation intensity was the highest during the same period. These meteorological factors contributed to the highest $O_3$ concentrations observed in 2019. In contrast, from May to October 2020, most regions in China experienced more precipitation, weaker radiation, and lower average maximum temperatures, creating meteorological conditions that were overall favorable for reducing $O_3$ concentrations (China Meteorological Administration, 2021).

4. Line 85: "time interval of 1 h" do you mean a temporal resolution of 1 h, or your data is for only a 1 hour interval?

**Response:**

Sorry for this confusion. "Time interval of 1 h" denotes a temporal resolution of 1 hour. In the revised manuscript, we have replaced "time interval of 1 h" with "temporal resolution of 1 h".

5. Line 93: "duration of $O_3$ pollution," please elaborate.

**Response:**

Thanks. It should be "duration of $O_3$ pollution episode", specifically talking about the

number of days during which $O_3$ concentration continuously exceed air quality standards.

6. Line 93: "can be divided into consecutive $O_3$-exceeding days with four or more days…" - please explain why this particular division was used?

**Response:**

This division was based on the features in Fig. 3. There was no discernible trend in the sum of days of episodes with less than four consecutive days in the three regions from 2015 to 2020. The upward trends are primarily controlled by consecutive O3-exceeding episodes lasting four days or more in all three regions from 2015 to 2020.

7. Line 117: The decrease in 2020 suggests correlation with emissions…

**Response:**

In line 117, we simply provide a brief description of the number of exceedance days and $O_3$ concentrations for each year, without delving into the analysis of the underlying causes at this point. Based on our subsequent analysis, the decrease observed in 2020 was primarily attributed to changing meteorological conditions.

8. Line 118: For completeness, please define "p" before use.

**Response:**

Thanks. Since p-value rather than "p" is a well-known terminology in statistical analysis, we have changed to "p-value" instead of "p" in the revised manuscript.

9. Line 127: "Is it due to changing $O_3$ photochemical processes or changing meteorological parameters?" – still the big question!

**Response:**

Yes, indeed! We also noticed that this sentence would be clearer if it is changed to "Is

it due to changing emissions of air pollutants or changing meteorological parameters?"
We have made the change in the revised manuscript.

10. Line 137: How does this expansion correlate with the expansion of urban/industrial regions to not previously developed regions (perhaps industries were relocated to surrounding regions from urban centerers?)

**Response:**

Please refer to our response to your general comment #2.

The expansion has been addressed specifically as follows: "We propose that the $O_3$ concentrations at the high $O_3$ stations stayed close to a saturation level of about 100 ppb throughout 2015 to 2020, even under downdrafts and stable atmospheric conditions brought about by mid-distance category TCs, was the result of a relatively high rates of atmospheric dispersion, dry deposition, and photochemical loss at high $O_3$ concentrations. This proposal is supported by modeling results (Li et al., 2012; Ouyang et al., 2022; Zhang et al., 2023). It is also consistent with theoretical consideration. While the low $O_3$ stations, where $O_3$ production were relatively small in 2015, experienced significant enhancements in the $O_3$ production (32 ppb in BTH, 12 ppb in PRD) from 2015 to 2017 because in 2017 the downdrafts and stable atmospheric conditions, which were highly conducive to $O_3$ formation, became more extensive due to changes in TCs and WPSH in the northwestern Pacific (Hu W. et al., 2023)."

11. Lines 141 to 144: Why compare the entire 2015-2020 period for high $O_3$ stations to the sub-period 2015-2017 for low $O_3$ stations?

**Response:**

Because "$O_3$ concentrations at the low $O_3$ stations caught up within 12 ppb with other stations in merely two years (an increase of about 30 ppb from 2015 to 2017), and actually equaled the average of other stations in 2019" as stated around lines 132 to 134 in the original manuscript.

12. Line 144: "quasi-saturation" – define.

**Response:**

Because both reviewers question the word "quasi-saturation", we have changed it to simply "saturation".

13. Line 145: "approximately 100 ppb" - what are the instrumental/measurement limitations for these sites?

**Response:**

According to "Environmental protection standards of the People's Republic of China Ambient air – Automatic determination of ozone – Chemiluminescence method (HJ 1225-2021)", the instrumental/measurement detection limit for $O_3$ is 0–500 ppb.

14. Line 147: "Did it have anything to do with the increase of consecutive $O_3$-exceeding days" – correlation!

**Response:**

No. Our analysis indeed began with correlation analysis. Once we observed correlations between different variables, we then seek theoretical support to explain this correlation and succeeded in most cases (Please refer to our response to your general comment #2).

15. Line 154: "expanded by about a factor of five from 2015 to 2017." - how did the land use/development change during this period?

**Response:**

Please see our response to your general comment #3.

16. Line 155: how big in area is the BTH box? How does it compare to the resolution of the data you analyzed?

**Response:**

The BTH box encompasses 218000 km$^2$. The spatial resolution of TAP data is 10 km, approximate 2180 grids used in the BTH box.

17. Line 156: "66.42 ppb in 2015 (31 days, Fig. 5a) to 69.44 in 2017 (62 days, Fig. 5b)" - comparing averages over different time periods (and number of days), how do you justify this comparison?

**Response:**

We apologize if this has caused any confusion. To the best of our knowledge, comparing O$_3$ pollution between different years, whether it's comparing O$_3$ exceedance concentrations or the number of exceedance days, is a common analytical approach. Regarding O$_3$ exceedance concentrations, the difference between 2017 and 2015 was 3.02 ppb. In terms of the number of O$_3$ exceedance days, there were 31 more days of O$_3$ exceedance in 2017 compared to 2015. We believe that this comparative approach is reasonable.

Furthermore, we consider both the number of O$_3$ exceedance days and O$_3$ exceedance concentrations as they together constitute the intensity of O$_3$ exceedance. This is why we use normalized O$_3$ concentrations to reveal the contribution of O$_3$ exceedance conditions to the mean O$_3$ concentration of the entire year.

18. Line 159: Not clear what the equation represents. Consider labeling with variables and defining the equation prior to usage….

**Response:**

We have replaced all calculation formulas in the revised manuscript with textual explanations.

19. Line 162: "driven primarily by the increase of consecutive O$_3$-exceeding days" – correlation!

**Response:**

We arrived at this conclusion by comparing the contributions of $O_3$ exceedance concentrations and the contributions of the number of $O_3$ exceedance days, rather than through correlation analysis. Please refer to our response to your general comment #2.

20. Line 163: "a lion's share…" – consider revising the wording!

**Response:**

We have changed "a lion's share" into "a predominant portion" in the revised manuscript.

21. Line 165: What do you mean by quasi-saturation? How does it work? What is the mechanism preventing further increase in concentrations? Related to measurement limitations at the stations?

**Response:**

In this study, "quasi-saturation" refers to the phenomenon that $O_3$ concentration becomes nearly saturated and stops increasing after reaching certain level (approximate 100 ppb). The next two questions are addressed in our response to your specific comment #10.

On the question "Related to measurement limitations at the station?" Environmental protection standards of the People's Republic of China Ambient air – Automatic determination of ozone – Chemiluminescence method (HJ 1225-2021)", the instrumental/measurement detection limit for $O_3$ is 0–500 ppb, which has nothing to do with the quasi-saturation phenomenon.

22. Line 166: "suggested that there was a quasi-saturation of $O_3$ inside Beijing City, and an expansion of weather systems conducive to $O_3$ formation from Beijing toward the southwest of the BTH box during 2017" - So the weather systems conducive to $O_3$ formation were previously focused on urban centers and now it has expanded to surrounding regions?! Please comment.

**Response:**

Thank you for a very insightful comment. Please refer to our response to your general comment #2. In particular, Fig. 13 illustrates for BTH "the weather systems conducive to $O_3$ formation were previously focused on urban centers and now it has expanded to surrounding regions?!".

23. Line 177: "Since it is highly unlikely" – please explain with data (e.g., emissions, land use/development) why it is highly unlikely!

**Response:**

Reviewer 1 had the same concern. We address this concern in the revised manuscript by consolidating and elaborating statements in "lines 166–168, 178–180, and many others" into a new section in line 243 as shown below.

**"3.3.1 Changing emissions as a possible cause of $O_3$ trends in 2015–2020**

As mentioned earlier, two emission-oriented hypotheses have been proposed as a possible cause of the $O_3$ trends in 2015–2020. One is changing emissions of $O_3$ precursors NOx and VOC (Li et al., 2022). The other is the reduced removal of $HO_2$ radicals by diminishing $PM_{2.5}$ suggested by Li K. et al. (2021) and Shao et al. (2021). Li et al. (2022) demonstrated convincingly that the NO titration effect was the cause of the linear trend in $O_3$ in PRD (0.5 ppb $yr^{-1}$) during the period 2006–2019. But for the period 2015–2020, the NO titration effect could account for only about 10% of the linear trend in $O_3$ of the low $O_3$ stations in PRD (5.0 ppb $yr^{-1}$, green line, Fig. S3a).

The increase of 30 ppb in $O_3$ at the low $O_3$ stations in BTH from 2015 to 2017 (green line, Figs. 4a and 8a represents about 50% increase in $O_3$. The titration effect can account for only about 5% (Fig. 8f). If this increase of 30 ppb in $O_3$ were due to an enhancement in $O_3$ precursors, the enhancement would have to be substantially greater than 50% because of the well-known less-than-linear relationship between changes in $O_3$ and its precursors, i.e., substantially more percentage changes in precursors are

needed for each percentage change in $O_3$ (Dodge, 1977; Shafer and Seinfeld, 1985). Figs. 8d and 8f show that CO (a proxy for VOC) and NOx changed only by a few percent from 2015 to 2017, more than one order of magnitude less than the changes needed. Hence it appears that changes in meteorological conditions conducive to $O_3$ formation are more likely the major contributing factor to the 50% increase in $O_3$ at the low $O_3$ stations in BTH. Similar argument can be extended to YRD and PRD (Figs. S1 and S3).

The theory of reduced removal of $HO_2$ radicals by diminishing $PM_{2.5}$ (25%, green line of Fig. 8c) appeared to be valid qualitatively for the 50% increase in $O_3$ at the low $O_3$ stations in BTH from 2015 to 2017 (green line of Fig. 8a). But this theory was contradicted directly by the phenomenon at the high $O_3$ stations where a 30% reduction in $PM_{2.5}$ (red line of Fig. 8c) corresponded to a decrease rather than an increase in $O_3$ (red line of Fig. 8a)."

[Figure]

**Figure S1.** Annual mean concentrations of maximum daily 8-hour average $O_3$ in YRD during $O_3$-exceeding days for all stations (black), high $O_3$ stations (red) and low $O_3$ stations (green) (a), same as (a) except for Ox (b), $PM_{2.5}$ (c), CO (d), $SO_2$ (e), $NO_2$ (f).

[Figure]

**Figure S3.** Annual mean concentrations of maximum daily 8-hour average $O_3$ in PRD during $O_3$-exceeding days for all stations (black), high $O_3$ stations (red) and low $O_3$ stations (green) (a), same as (a) except for Ox (b), $PM_{2.5}$ (c), CO (d), $SO_2$ (e), $NO_2$ (f).

24. Line 249: "into two groups" – why these two groups?

**Response:**

Please see our response to your specific comment #6.

25. Line 327: include full forms in the section titles rather than acronyms.

**Response:**

Done accordingly.

26. Line 320-323: So, the increased ozone is due to reduction in removing processes (low advection and low mixing) while the production is the same and not increased?

**Response:**

The downdrafts and stable atmospheric conditions are usually associated with meteorological conditions of clear skies and high surface temperatures which are highly

conducive to higher photochemical production as well as reduced "removing processes (low advection and low mixing)" of $O_3$. Please refer to our response to your general comment #2.

27. Line 343: "… in the former" - do you mean $O_3$ exceeding days?

Yes, "… in the former" means $O_3$-exceeding days. To avoid confusion, we have replaced the former with "$O_3$ exceeding days", and the latter with "clean days" in the revised manuscript.

28. Please comment on your choice for compare the average conditions over (for example) 31 days for $O_3$ exceeding days to average over 152 clear days? What happens if you compare $O_3$ exceeding days to average over the entire period including both clear days and $O_3$ exceeding days?

**Response:**

Comparing the average conditions over all $O_3$-exceeding days to the average over 152 clean days is aimed at providing an intuitive analysis of the differences in meteorological patterns between polluted and clean days.

Following your suggestion, we have also compared $O_3$ exceeding days to average over the entire period including both clean days and $O_3$ exceeding days, as shown in Fig. S11. It is clear that this additional comparison does not affect the validity of our conclusions.

29. Fig 12: what does the concentration in ppb in parentheses refer to? Figure details are not clear (e.g., isoline labels are hard to read, 5880 gpm is not even labeled)

**Response:**

The concentration in ppb in parentheses refer to average $O_3$ concentration. We have revised all the figures in the revised manuscript.

Technical corrections:

1. I suggest using different markers in Figures 1,2, and 4, so that different curves are discernible on a grayscale (black and white) version of your manuscript.

**Response:**

Done accordingly.

2. Table 3, 4: write complete captions rather than "same as…"

**Response:**

Done accordingly.

3. Figures 5,6: complete the captions.

**Response:**

Done accordingly.

4. Line 478 and 492: hyperlinks don't work, please check!

**Response:**

Checked.

---

## Author Response (AR2)

Dear Editor,

We have carefully considered all comments and suggestions. Listed below are our point-by-point responses to all comments and suggestions of two referees (Reviewer's points in black, our responses in blue).

In response to a key comment of Referee #1, we decided to elaborate in the revised manuscript on the saturation at high-concentration sites by adding a new section "3.3.4 Saturation at high-concentration sites" plus a new figure (Figure 16), to which Shanshan Ouyang had contributed significantly. We hereby inform and explain that to you.

**Referee #1**

The revision has partly addressed my concerns. The clarity of sections 3.1 and 3.2 has been largely improved. However, Section 3.3 is still not convincing and does not show sufficient novelty to meet ACP standard. It may require another round of substantial revision. My comments are to the revision-tracked version.

**Response:**

Instead of disputing the level of novelty of this study, we would like to point out that this study is the first (i.e., original) to suggest that "changes in meteorological conditions" rather than "changes in emissions" are the cause of of positive ozone trends in the three megacity clusters in eastern China during 2015–2020. We believe that the level of originality is a critical merit for publishing a paper. Moreover, the prevailing view on the cause is "changing emissions", this study is trying to change that view, and we believe the balance of evidence is in our favor.

Regarding Section 3.3, we understood your concern and decided to elaborate on the saturation at high-concentration sites in a new section "3.3.4 Saturation at high-concentration sites" plus a new figure (Figure 16). Please see our response on your comment "Line 294-305: Again, why this only causes ozone increase in low-ozone sites, but not at high-ozone sites? It looks like it is an important finding from the observations

but no convincing reasons are provided."

I suggest use "variability" other than "trends" in the title and the text, because 2015-2020 are too short to derive statistically meaningful trends (only 6 data points). It also makes much more sense if the authors prefer to highlight meteorological conditions of 2017 and 2019 in their analysis.

**Response:**

Changing "trends" to "variability" for the short term trend a logical suggestion. We accept this suggestion, but would like to point out that all previous "emission-caused trends" papers used the term "trends" for the same short period. Moreover, we have to change most of the "trends" in the text to "increases" rather than "increased variabilities" which could have a misleading meaning.

Line 98: may rephrase to "some commonly-used methods are applied in this study."

**Response:**

Thanks! Done accordingly.

Line 140-146: It is not necessary that high ozone concentrations are at urban centers, as titration would be strong at urban centers which lead to lower ozone compared to suburban regions. Please provide clear information to support this statement.

**Response:**

The two figures below show that, despite the titration effect, Beijing and Shanghai both have higher ozone concentrations compared to their corresponding suburban regions. So is in PRD (Figure 7, not shown).

[Figure]

Figure 5: Spatial distribution of annual mean concentrations of maximum daily 8-hour average $O_3$ for $O_3$-exceeding days in BTH in 2015 (a), 2017 (b) and their difference (2017–2015) (c). The top, middle and bottom rectangle boxes denote BTH, YRD and PRD districts, respectively. The number inside the parenthesis behind 2015 or 2017 denotes the number of $O_3$-exceeding days.

[Figure]

Figure 6: Spatial distribution of annual mean concentrations of maximum daily 8-hour average $O_3$ for $O_3$-exceeding days in YRD in 2015 (a), 2017 (b) and their difference (2017–2015) (c). The top, middle and bottom rectangle boxes denote BTH, YRD and PRD districts, respectively. The number inside the parenthesis behind 2015 or 2017

denotes the number of $O_3$-exceeding days.

Line 151: Has Figures 5 been introduced before? I suggest show raw site measurement instead of the gridded data because the later may introduced unrealistic smooths.

**Response:**

Figure 5 has not been introduced before. Gridded data of MDA8 $O_3$ in 2015–2017 were reported by Xue et al. (2020). They found that ozone site measurement data correlated well with ozone grid data. Furthermore, we found that gridded data used in Figures 5, 6 and 7 provided a better nationwide perspective of $O_3$ distributions and variabilities than those from site data.

Line 206-207. Many other studies have suggested this hypothesis. Suggest remove "suggested by XXX", instead just use them as references.

**Response:**

Agree. Done accordingly.

Line 208-209: I suggest tune down this statement. It is only a possible and partial cause.

**Response:**

Done as suggested.

Section 3.3: Could the title of 3.3 be something like "Causes of ozone expansion at low-concentration sites and saturation at high-concentration sites". I am still not clear whether "saturation" is a precise word to describe the "ozone remained nearly constant".

**Response:**

Title of 3.3 has been changed to "Causes of ozone enhancement at low-concentration sites and saturation at high-concentration sites".

Line 225-262: Suggest shorten this part. It takes too long to start the discussion of

meteorology as stated by the subtitle.

**Response:**

Thanks! We have shortened line 225–262 by about 50% in the revised manuscript as shown below:

"Fig. 9a shows the mean daily $O_3$ concentrations of the first group with four or more consecutive $O_3$-exceeding days (labeled $O_3$ days≥4) in 2015, Fig. 9b shows the mean daily $O_3$ concentrations of the second group with less than four consecutive $O_3$-exceeding days (labeled $O_3$ days<4), and Fig. 9c is the difference between the two groups (6.10 ppb, Table 2). Figs. 9d–9f are the same as Figs. 9a–9c, respectively, but for 2017. The first group in 2017 had 28 days and mean $O_3$ of 74.43 ppb inside the BTH box, while the second group had 34 days and 65.32 ppb (Table 2). One of the most remarkable differences between 2017 and 2015 in Figs. 9a–9f was the large number of days with four or more consecutive $O_3$-exceeding days (first group) in 2017 (28 days, Fig. 9d) over that of 2015 (7 days, Fig. 9a), which alone contributed to about 62% of the difference in $O_3$ between 2017 and 2015 as shown in Fig. 2a (red line). Approximately 30% was contributed by the 10 days' difference (2017 vs. 2015) in the number of days with less than four consecutive $O_3$-exceeding days (second group). The contribution by the higher average concentration of MDA8 $O_3$ of the first group in 2017 is only about 8% (Table 2). These values of contributions reconfirm what is shown in Fig. 3a, i.e., the greater frequency of episodes with four or more consecutive $O_3$-exceeding days contributes the majority (62%) to the higher $O_3$ in BTH in 2017 vs. 2015, the greater intensity/concentration of $O_3$ during the episodes contributes only about 8%, consistent with the expansion and saturation effect discussed earlier.

The phenomena illustrated in Figs. 9a–9f also exist in YRD and PRD as well as in most other years. Figures equivalent to Figs. 9a–9c for all years in the three city clusters are provided in the Supplementary Material (Figs. S4–S6). Essential information derived from those figures is summarized in Tables 2–4.

In Figs. 10a and 10b the values of SSR and T2m of the episodes with four or more consecutive $O_3$-exceeding days are compared to those of $O_3$ episodes with less than four consecutive $O_3$-exceeding days, and to those of clean days (non-$O_3$-exceeding days). As expected, the $O_3$ episodes with four or more consecutive $O_3$-exceeding days consistently have the highest values of SSR and T2m, while the clean days have the lowest values. This is the case in nearly all years studied as shown in the Supplementary Material (Fig. S7) and is also generally true in YRD and PRD (Figs. S8 and S9)."

Line 285-294: It is still very difficult to understand what exactly explain the "saturation" at high-ozone sites. Why "This saturation effect was the result of enhanced rates of atmospheric dispersion, dry deposition and photochemical loss at high $O_3$ concentrations"? Please clarify with data supported.

**Response:**

Since this comment and the next deal with the same issue, please see our response below.

Line 294-305: Again, why this only causes ozone increase in low-ozone sites, but not at high-ozone sites? It looks like it is an important finding from the observations but no convincing reasons are provided.

**Response:**

We appreciate very much this insightful comment and the one above. Actually you raised the same point in the earlier round of review. The spatial expansion and saturation of high $O_3$ was regarded by us as an interesting empirical finding, your persistent comments make us realize that this finding may have more important and far reaching implications than previously perceived. So we decided to elaborate on the saturation at high-concentration sites by adding a new section "3.3.4 Saturation at high-concentration sites" plus a new figure (Figure 16) which are shown below.

"**3.3.4 Saturation at high-concentration sites**

Why the favorable meteorological conditions only cause $O_3$ increase at low $O_3$ stations, but not at high $O_3$ stations? And why the saturation $O_3$ level is around 100 ppb as shown in Fig. 4? These questions can be best addressed by examining Fig. 16 which depicts the time series of individual processes (where DDEP denotes dry deposition, CHEM the net photochemical production of $O_3$, HTRA the horizontal transport and VTRA the vertical transport) contributing to $O_3$ budget in PRD (averaged over 56 stations in PRD) calculated by the WRF-CMAQ model for the $O_3$ episode of September 24–October 1, 2019 (Ouyang et al., 2022). This episode was one of the most sever $O_3$ episodes since the official $O_3$ observation started in PRD in 2006. MDA8 $O_3$ exceeded the 75 ppb standard on all eight days of the episode. Hourly $O_3$ reached as high as 110 ppb, yet all MDA8 $O_3$ stayed approximately between 75 and 100 ppb. This suggests a ceiling/saturation level of approximately 100 ppb for MDA8 $O_3$, consistent with what was observed in Fig. 4 for PRD as well as BTH and YRD. Since this episode was one of the most sever episodes, we can assume that the 100 ppb saturation level would also be applicable to other $O_3$ episodes in Guangdong. More importantly, the saturation effect was also a common feature in the results of other three-dimensional models for other megacity clusters, in which MDA8 $O_3$ usually saturated at 100 ppb, e.g., in YRD (Li et al., 2012) and in Beijing (Zhang et al., 2023). This explains why the saturation $O_3$ level is around 100 ppb as shown in Fig. 4.

In regard to the first question: Why the favorable meteorological conditions only cause $O_3$ increase at low $O_3$ stations, but not at high $O_3$ stations? It can be understood as follows: At a low $O_3$ station of 65 ppb MDA8 $O_3$ in PRD in 2015 (Fig. 4c), Fig. 16 shows that MDA8 $O_3$ can readily increase to 75–100 ppb in a few hours from an early morning low ozone of about 50 ppb under favorable meteorological conditions. However, at a high $O_3$ station of 100 ppb MDA8 $O_3$ in 2015 (Fig. 4c) under the same favorable meteorological conditions, MDA8 $O_3$ would also reach 75–100 ppb in a few hours from an early morning low ozone of about 50 ppb (Note here we assume all stations start the day with an early morning low ozone of 50 ppb, consistent with the value in Fig. 16). In other words, the saturation levels at all stations are the same at 75–

100 ppb, independent of the ozone concentration in 2015.

In terms of contributing processes, the saturation level of 75–100 ppb is controlled primarily by photochemical loss, dry deposition and dispersion to the free troposphere. This can be clearly seen in Fig. 16, on all eight days in the mid-morning when $O_3$ is approaching toward its peak value, CHEM declines sharply due to photochemical loss, and HTRA, VTRA and DDEP all become greater. Near noontime $O_3$ starts to drop sharply."

[Figure]

Figure 16: Time series of individual processes contributing to $O_3$ budget in PRD calculated by the WRF-CMAQ model for the $O_3$ episode of September 24–October 1, 2019. The black line ($O_3$) represents the averaged $O_3$ concentrations in the layers below 1260m. Where DDEP denotes dry deposition, CHEM denotes chemical processes, HTRA denotes the horizontal transport and VTRA denotes the vertical transport.

Finally, in the following figure the data of 2021 and 2022 have been added to Figure 4 and presented in the revised Supplementary as Figure S14. As you can see that the saturation effect remains intact (red lines stay lower than 100 ppb), while green lines stay significantly higher than those in 2015.

[Figure]

Figure S14. Annual mean concentrations of maximum daily 8-hour average $O_3$ during $O_3$-exceeding days for all stations (black), high $O_3$ stations (red) and low $O_3$ stations (green) in BTH in 2015–2022 (a), YRD (b) and PRD (c).

My judgement is that the study does not add significant novelty to the meteorological cause of high ozone in 2017 and 2019, because the impact of tropical cyclone and subtropical high is well-known. This is can be seen in Sections 3.3.2-3.3.3 that the study cites and repeats many existing research. Please highlight what's novel in this analysis. Why this happens in 2017 and 2019? Why this extends the ozone episodes? Can such weather pattern explain the observed ozone increase in low-ozone site and the saturation of high-ozone site?

**Response:**

Please refer to our response to your comment about the novelty at the start of this review. Please also refer to our response to your comment on the saturation of high-ozone sites around line 294–305. In addition, while "the impact of tropical cyclone and subtropical high is well-known", we deserve the credit of being the first to apply this well-known information to argue for "changes in meteorological conditions" rather than "changes in emissions" being the cause of positive ozone trends in three megacity clusters in

eastern China during 2015–2020.

Figure 2. I am confused about the y-axis title of Figure 2. Ozone days has lower than 20 ppbv of MDA8 ozone? Please check carefully.

**Response:**

Line 99–101, we stated "The normalized annual mean $O_3$ concentration of the $O_3$-exceeding days is calculated by adding the $O_3$ concentration of the $O_3$-exceeding day each year and dividing it by the total number of days in the year. The normalized annual mean $O_3$ of the non-$O_3$-exceeding days is calculated by the same method except for the non-$O_3$-exceeding days."

**References**

Xue, T., Zheng, Y., Geng, G., Xiao, Q., Meng, X., Wang, M., Li, X., Wu, N., and Zhang, Q., Zhu, T.: Estimating spatiotemporal variation in ambient ozone exposure during 2013–2017 using a data-fusion model, Environ Sci Technol, 54, 23, 14877−14888, https://doi.org/10.1021/acs.est.0c03098, 2020.

**Referee #2**

I couldn't find any reference to Table R1 and Figures R1 to R3 (in Author's response document) or the corresponding information in either the main manuscript or the supplement? Please include at least some of this information in the manuscript.

Response:

Sorry for the confusion. We did put Figures R1 and R2 (where R denotes Response) in the supplementary material but rename them as Figures S1 and S3 (where S denotes Supplementary). Figure R3 was not included in the supplementary material because it was not cited in the text of the main manuscript.

Also, the following paragraph as added text in Author's response doesn't appear in Author's tracked changes. I wonder why that is?

"Hu W. et al. (2023), in collaboration with this study, conducted a statistical analysis to assess processes that contribute to high $O_3$ formation in PRD when TCs were present in the northwest Pacific. They investigated the impact of the distance between TCs in the northwest Pacific and PRD on the $O_3$ concentration in the PRD from 2006 to 2020. They found that the large numbers of consecutive $O_3$-exceeding days in 2017 and 2019 relative to 2015 were primarily attributable to the greater occurrence of downdrafts and stable atmospheric conditions brought about by mid-distance category TCs. This finding clearly establishes that changing frequency of mid-distance category TCs (i.e. changing meteorological conditions) is the cause of the increases in the numbers of consecutive $O_3$-exceeding days as well as the higher $O_3$ concentrations in PRD. Ongoing study by our research group further shows that the mid-distance category TCs are predominately those TCs with tracks starting around the southern Philippines and ending near Korea and/or Japan. Since TC tracks in northwestern Pacific are strongly controlled by WPSH, we conclude that both Philippines-to-Korea/Japan track TCs and corresponding distribution and intensity of WPSH contributed to the higher consecutive $O_3$-exceeding days in PRD from 2015 to 2020."

**Response:**

Please look for this paragraph around line 265–276 in the current version of the manuscript. It was around line 282–292 in the earlier version.